# An essential role for MEF2C in the cortical response to loss of sleep in mice

Theresa E Bjorness[1,2†], Ashwinikumar Kulkarni[3†], Volodymyr Rybalchenko[1†], Ayako Suzuki[1], Catherine Bridges[4], Adam J Harrington[4], Christopher W Cowan[4], Joseph S Takahashi[3,5], Genevieve Konopka[3*], Robert W Greene[1,3,6*]

[1]Department of Psychiatry, Peter O'Donnell Brain Institute, University of Texas Southwestern Medical Center, Dallas, United States; [2]Research Service, North Texas VA Health Care System, Dallas, United States; [3]Department of Neuroscience, Peter O'Donnell Brain Institute, University of Texas Southwestern Medical Center, Dallas, United States; [4]Department of Neuroscience, Medical University of South Carolina, Charleston, United States; [5]Howard Hughes Medical Institute, University of Texas Southwestern Medical Center, Dallas, United States; [6]International Institute of Integrative Sleep Medicine, University of Tsukuba, Tsukuba, Japan

*For correspondence:
genevieve.konopka@
utsouthwestern.edu (GK);
robertw.greene@utsouthwestern.
edu (RWG)

[†]These authors contributed
equally to this work

Competing interest: See
page 22

Reviewing editor: Anne E West,
Duke University School of
Medicine, United States

**Abstract** Neuronal activity and gene expression in response to the loss of sleep can provide a window into the enigma of sleep function. Sleep loss is associated with brain differential gene expression, an increase in pyramidal cell mEPSC frequency and amplitude, and a characteristic rebound and resolution of slow wave sleep-slow wave activity (SWS-SWA). However, the molecular mechanism(s) mediating the sleep-loss response are not well understood. We show that sleep-loss regulates MEF2C phosphorylation, a key mechanism regulating MEF2C transcriptional activity, and that MEF2C function in postnatal excitatory forebrain neurons is required for the biological events in response to sleep loss in C57BL/6J mice. These include altered gene expression, the increase and recovery of synaptic strength, and the rebound and resolution of SWS-SWA, which implicate MEF2C as an essential regulator of sleep function.

## Introduction

Sleep abnormalities are commonly observed in numerous neurological disorders, including autism spectrum disorder, major depressive disorder, bipolar disorder, post-traumatic stress disorder, neurodegenerative disorders and many others, but our understanding of sleep need and its regulation and resolution is poorly understood. Following an extended period of waking, or sleep deprivation (SD), the mammalian cortex shows an altered pattern of EEG activity characterized by rebound slow wave power (delta power in the frequency range of 0.5–4.5 Hz) during the ensuing slow wave sleep (SWS; also referred to as NREM) periods. The amplitude of the rebound directly correlates with the previous waking or SD duration (*Borbély, 1982*; *Franken et al., 2001*) implicating rebound SWS, slow wave activity (SWS-SWA) as a biomarker of sleep need. During SWS episodes, the rebound SWS-SWA increase resolves, consistent with resolution of sleep need (*Bjorness et al., 2016*).

The buildup and resolution of sleep need is correlated with overall cortical, excitatory synaptic strength. Based on sleep-related modulation of excitatory synapse biochemistry, morphology and electrophysiological activity, a scaling-up during waking and a scaling-down during sleep have been observed (*Bushey et al., 2010*; *de Vivo et al., 2017*; *Diering et al., 2017*; *Liu et al., 2010*), although this may not necessarily be reflected by the neuronal firing rates (*Hengen et al., 2016*; *Watson et al., 2016*).

Changes in synaptic transcript (*Noya et al., 2019*) and protein (*Diering et al., 2017*; *Noya et al., 2019*) expression, required for synaptic down-scaling, are observed during the sleep phase in

association with an average decrease in synapse size (*de Vivo et al., 2017*; *Diering et al., 2017*). Characterization of the cell-signaling biochemical and molecular mechanisms responsible for these sleep-related changes in protein expression is needed.

MEF2 transcription factors are critical regulators of activity-dependent synapse elimination (reviewed in *Assali et al., 2019*). MEF2C plays an essential role in development to control cortical synaptic connectivity, function and plasticity (*Harrington et al., 2016*; *Li et al., 2008*; *Rajkovich et al., 2017*). MEF2C's activation results in excitatory synapse down-regulation with potential relevance for sleep, in that prolonged waking activity leads to excitatory synaptic scaling-down during ensuing sleep, as noted above. Neuronal activity associated with waking, that may include neuronal depolarization, firing and arousal-associated neuromodulator activity, can switch MEF2C from a transcriptional repressor to an activator. The switch depends on dephosphorylation of a conserved Serine (S396) (*Kang et al., 2006*; *Lyons et al., 2012*; *Zhu and Gulick, 2004*), and both the repressor and activator functions of MEF2C appear critical for its regulation of synaptic transmission (*Harrington et al., 2016*). However, the relationship between MEF2C and sleep-regulated transcripts, sleep-regulated excitatory synaptic strength and sleep-homeostatic EEG activity remains unexplored.

In this study, we examined neurobiological changes in the mouse frontal cortex in response to sleep loss, involving transcriptomic architecture, regulation of synaptic strength in excitatory cortical neurons and SWS-SWA, correlated with sleep need. As changes in neuronal activity may be observed across sleep-wake states, we investigated the possible role of MEF2C as a key regulator of sleep-related control of gene expression, sleep-related modulation of synaptic function and sleep-need correlated SWS-SWA buildup and resolution.

## Results

### Wake-sleep mediated control of the transcriptome

To evaluate the effects of sleep loss, we compared three sleep-related conditions for differentially expressed genes (DEGs) from samples of the frontal cortex (FC, the cortical region showing the highest power of SWS-SWA) of male C57BL/6J (WT) mice (*Figure 1A*) including: (1) control sleep (CS; mice free to sleep, ad libitum; n = 6), (2) sleep deprived for 6 hr (SD; n = 7), and (3) sleep deprived for 4 hr followed by recovery sleep for 2 hr (RS; n = 7). All samples were collected 6 hr after lights on (ZT6) to control for circadian influence and were processed for RNA-seq analysis (see Materials and methods).

The CS samples expressed, on average, ~13,000 genes with counts per million reads (CPM) $\geq$1.0 in the FC. Comparing the SD samples to the CS samples, we observed 6,248 DEGs with roughly half the number of genes having decreased expression (FDR <= 0.05; |log$_2$ (fold change)|>=0.3; *Figure 1B*; *Supplementary file 1*), confirming an earlier report of massive transcriptome changes associated with SD (*Diessler et al., 2018*). Similarly, the RS cohort showed 3827 DEGs compared to CS, again with approximately half the number of genes showing decreased expression (*Figure 1C*; *Supplementary file 1*).

In contrast, only 81 genes showed significantly altered expression comparing the SD condition to RS (*Supplementary file 1*), suggesting that for the majority of SD-sensitive genes, RS is more than simply a direct recovery of altered gene expression back to CS levels. Rather, more than half of the differential expression initiated during SD is continued during RS (*Figure 1D*), significantly extending the SD-induced change.

Because SD may be considered stressful (*Mongrain et al., 2010*), we compared classic glucocorticoid-mediated response genes to the increased DEGs observed in response to SD (*Figure 1—figure supplement 1*, *Supplementary file 2*). No significant increase in glucocorticoid (GC) intracellular response genes was found in the upregulated SD DEGs (p-value=0.513 using Fisher's exact test), suggesting a more SD-specific response than other GC-mediated stress responses.

This bulk transcriptomic analysis includes mRNA from all cell types found in frontal cortical tissue. Many of the genes expressed in these tissues are, nonetheless, cell-type specific, allowing potential correlation between DEGs specific for a sleep condition and specific cell types. Using previously published single-cell RNA-seq from the adult mouse cortex (*Hrvatin et al., 2018*), we identified cell-type-specific gene expression patterns across sleep conditions (*Figure 1E*). Interestingly, the

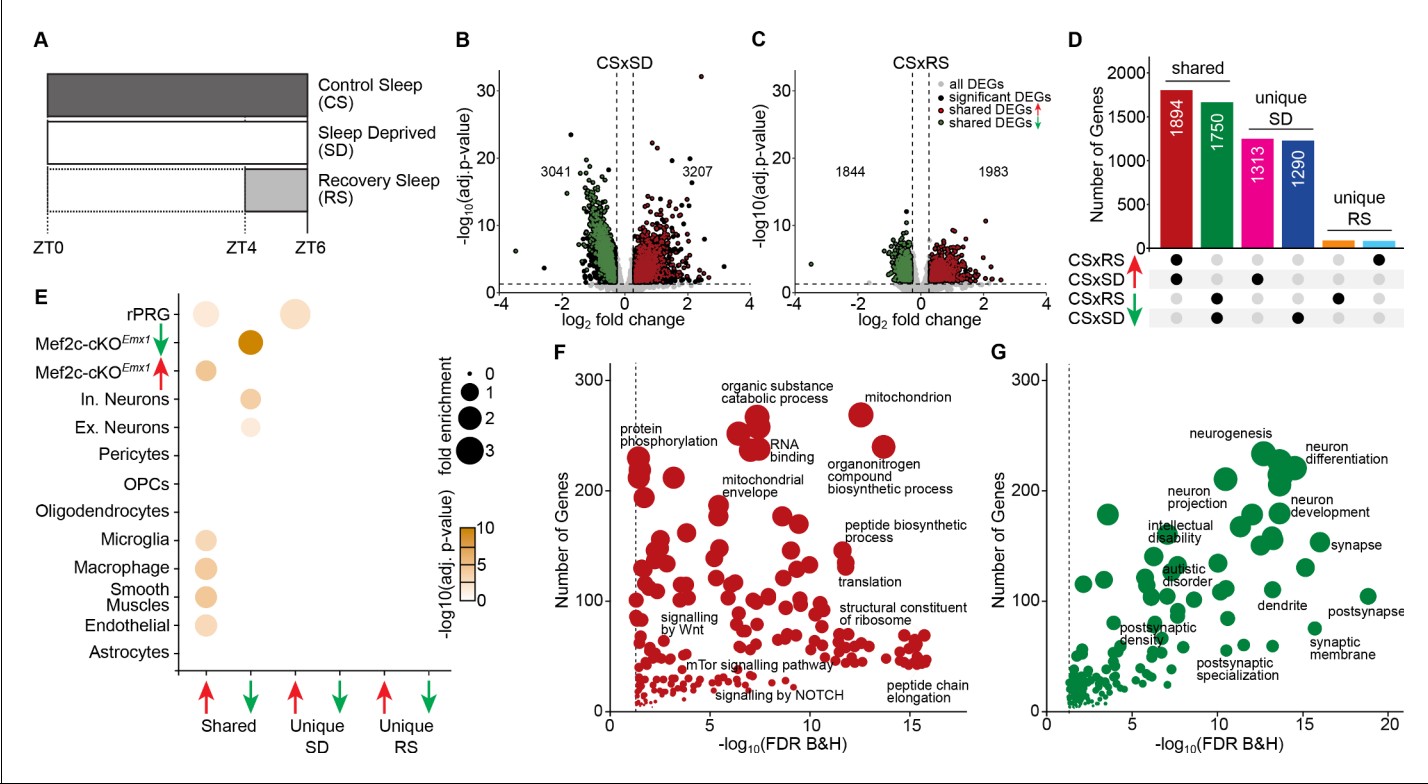

**Figure 1.** Sleep need induced transcriptomic changes. (**A**) Schematic of the experimental design illustrating the protocol for frontal cortex brain tissue collection from three sleep conditions at Zeitgeber Time (ZT) = 6 hours: control sleep (CS), sleep deprived (SD) and recovery sleep (RS). Volcano plots showing differentially expressed genes across (**B**) CS to SD and (**C**) CS to RS. Significantly differentially expressed genes (DEGs) (adj. p-value<=0.05, absolute log2 fold change >= 0.3), unique to the sleep condition, are indicated as *black* dots whereas genes shared by SD and RS are indicated as either *red* (increased expression) or *green* (decreased expression) dots. (**D**) Upset plot showing the shared and unique sets of significant DEGs for CS to SD and CS to RS. (**E**) Dot plot showing enrichment of significant DEGs for cell-types identified using single cell RNA sequencing data (*Hrvatin et al., 2018*) from mouse cortex, identified *Mef2c* target genes (*Harrington et al., 2016*) and rapid primary response genes (rPRGs) (*Tyssowski et al., 2018*). A Fisher's exact test was used to calculate the significance of the overlap of the sleep DEGs and the Mef2c-cKO$^{Emx1}$ DEGs. The size of the dot indicates the odds ratio for enrichment and the color indicates the p-value. Dot plot showing GO analysis for shared significant DEGs with (**F**) increased and (**G**) decreased expression across SD and RS compared to CS (for a complete list of significant GO, see *Supplementary file 3*).

The online version of this article includes the following figure supplement(s) for figure 1:

**Figure supplement 1.** 3207 genes increased their expression after 6 hr of sleep deprivation (SD).

**Figure supplement 2.** Kreb's cycle with highlighted differentially expressed genes with increased expression in both SD and RS, and involved in anaplerotic reactions.

**Figure supplement 3.** Circadian rhythm gene regulatory network with highlighted core clock genes that are affected during SD.

downregulated DEGs shared by SD and RS were enriched for genes associated with both excitatory and inhibitory neurons, whereas the upregulated DEGs shared by SD and RS were enriched for non-neuronal genes that are down in the Mef2c-cKO$^{Emx1}$ mice (*Harrington et al., 2016* ; *Figure 1E*).

We also examined MEF2C target gene expression in response to SD or RS. We found that the downregulated DEGs shared by SD and RS were enriched for neuronal genes affected by *Mef2c* loss of function as previously observed in a conditional knockout of pan-neuronal *Mef2c*, Mef2c-cKO$^{Emx1}$ (*Harrington et al., 2016*). Whereas the upregulated DEGs in response to SD and RS were enriched for *Mef2c* non-neuronal genes (*Figure 1E*). Thus, the DEGs altered by SD and RS in the cortex significantly overlap those altered by the loss of cortical MEF2C function.

We next functionally characterized the sleep-loss induced transcriptional changes shared by SD and RS states by carrying out a gene ontology (GO) analysis. Overall, there are distinct GO categories for the upregulated and downregulated genes (*Figure 1F,G*; *Supplementary file 2*). The upregulated categories include mitochondrial components and signaling pathways of oxidative phosphorylation and anaplerosis (*Figure 1—figure supplement 2*), indicative of energy mobilization,

as well as anabolic protein metabolism that includes ribosomal components together with transcriptional, translational and related processes. These results are consistent with and extend those from a previous study using microarrays (*Cirelli et al., 2004*) that lead to a conclusion, '... that sleep, far from being a quiescent state of global inactivity, may actively favor specific cellular functions'. Recently, similar findings were reported in a RNA-seq analysis from C57BL/6J whole cortex and a 'gentle handling' method of SD (*Hor et al., 2019*).

Several studies provide converging evidence for a general decrease in cortical excitatory synaptic strength in association with sleep (*de Vivo et al., 2017*; *Liu et al., 2010*; *Maret et al., 2011*) with respect to protein expression in synaptosomes (*Diering et al., 2017*) and spontaneous miniature excitatory post-synaptic currents and frequency (*Liu et al., 2010*; *Vyazovskiy et al., 2008*). Consistent with this, we observed that the downregulated genes observed in SD and RS are enriched for GO categories reflecting synaptic biological processes and cellular components. These genes are also enriched for genes involved in diseases of cognition associated with abnormal synaptic function, including schizophrenia, autism, intellectual disability and neurodegenerative disorders (*Figure 1G*, *Supplementary file 3*).

Some of the SD upregulated DEGs have a notable role as both transcription factors and/or first responders to neural activity and sensory stimulation. Rapid, primary response genes (rPRG) have been identified in cortical neuronal cultures as responders to both brief or 6 hr exposures to a depolarizing concentration of potassium in the presence of a translational blocker (*Tyssowski et al., 2018*). We find that DEGs for CS compared to SD and/or RS conditions are significantly enriched for these rPRGs (*Figure 1E*). Furthermore, upregulated transcription factors were identified in visual cortex that responded to a short exposure to light after being in constant darkness (*Hrvatin et al., 2018*). We observed that 50% of these were upregulated DEGs for CS to SD (listed in *Supplementary file 1*). Notably, these upregulated, transcription factors in the visual cortex overlapped with DEGs from the FC, and showed sustained, increased expression even after 6 hr of SD-induced activation.

It was recently shown that the influence of circadian rhythms on gene transcript expression in synaptosomes was relatively unaffected by SD, although SD did prevent translation (*Noya et al., 2019*). Nevertheless, many core clock genes are known to affect sleep homeostasis and the expression of some core clock genes, in particular, *Per2*, are altered by SD, as previously reviewed (*Franken, 2013*). To control for circadian-mediated changes in gene expression, we assessed DEGs at the same circadian time, ZT6, changing only the 6 hr, immediate sleep history. In contrast to DEGs in synaptosomes, we observed in FC tissue that SD from ZT = 0 to 6 hr resulted in a number of core clock DEGs (compared to CS; *Figure 1—figure supplement 3*), including downregulation of *Arntl*, *Clock* and *Npas2* and downregulation of a set of genes regulating circadian entrainment (MGI; GO:0009649). Taken together, these data are consistent with an SD-induced attenuation and destabilization of circadian-mediated gene expression and confirm recent observations using an RNA-seq assessment in time series together with SD (*Hor et al., 2019*).

To further uncover the organizational architecture and prioritize the extensive SD-mediated transcriptome, we used weighted gene co-expression network analysis (WGCNA), with expressed genes from all three conditions (CS, SD, and RS) to generate a network of 59 modules. Many of the modules (53%) were enriched for either up or downregulated DEGs from CS to SD conditions (*Figure 2A*). Most of those enriched for upregulation from CS to SD (n = 11), were also enriched for upregulation from CS to RS (n = 10), and similarly, most modules enriched for downregulation from CS to SD (n = 18) were also enriched for downregulation from CS to RS (n = 13), showing an organization primarily driven by DEGs, that is consistent between SD and RS conditions. The GO of these shared modules (*Supplementary file 4*) is also consistent with those seen using only the lists of DEGs.

The advantage of WGCNA is that it allows for the prioritization of gene modules that may be central drivers of biological networks independent of arbitrary cutoffs of p-values and fold changes for individual genes. WGCNA identifies these modules containing genes with highly correlated expression that can be further assessed for significant expression patterns related to features such as the sleep conditions of CS, SD and RS via determination of module eigengenes.

Notably, putative target genes of MEF2C, were enriched in a surprising, 15 of 59 modules generated from the three sleep conditions employed in this study. Using a Monte Carlo model to assess the likelihood of such a result gave a probability of p<0.001 from a sample of 1000 permutations of

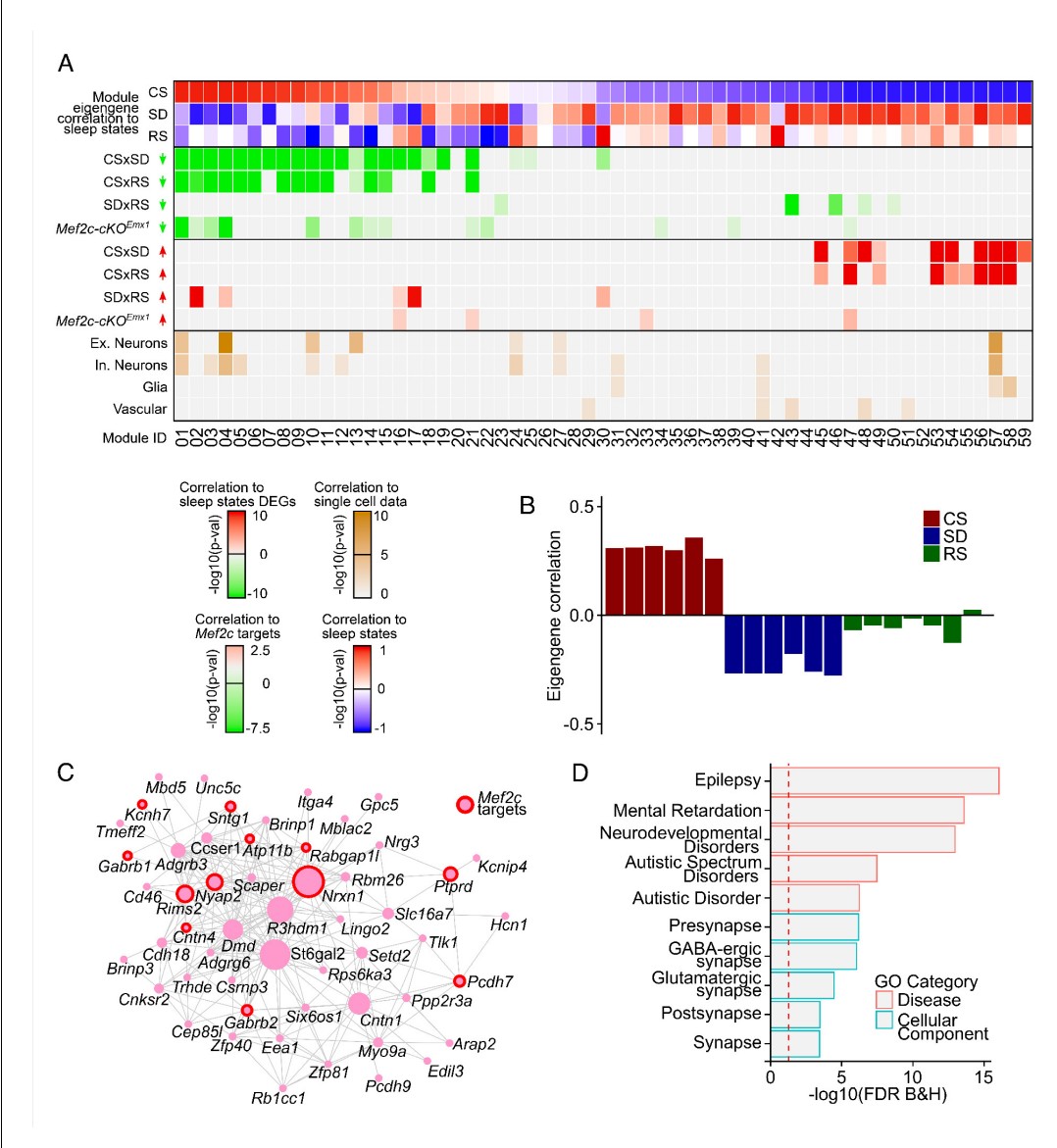

**Figure 2.** Coordinated transcriptomic responses with sleep need indicate a role for MEF2C. (A) Heatmap showing enrichment of sleep-state DEGs across modules of genes identified by weighted gene co-expression network analysis (WGCNA). First three rows show module eigengene correlation to sleep states (CS, SD and RS respectively) ordered by most positive to most negative correlation to CS sleep state. The following four rows highlight the significance of module enrichment for a specific set of DEGs colored by *green* for genes with decreased expression for CSxSD, CSxRS, SDxRS and MEF2C targets (*Harrington et al., 2016*) and the next four rows colored by *red* to highlight the significance of module enrichment for genes with increased expression for CSxSD, CSxRS, SDxRS and MEF2C targets (*Harrington et al., 2016*). The bottom four rows colored by shades of *orangeyellow* show enrichment of modules for cell types identified using single-cell RNA sequencing (*Hrvatin et al., 2018*) from mouse cortex. Each column is a module of genes as indicated by the module ID in last row. (B) Barplot for a selected module (Module ID 04) with eigengene correlation across sleep states showing a strong positive correlation to CS and negative correlation to SD and weak negative correlation to RS. (C) Network plot for selected module (Module ID 04) showing the top 250 connections between genes. Genes that are MEF2C targets (*Harrington et al., 2016*) are highlighted by red circles. (D) GO plot for genes within the selected module (Module ID 04).

The online version of this article includes the following figure supplement(s) for figure 2:

**Figure supplement 1.** A module of gene co-expression characterized by an eigengene driven by sleep condition.

genes randomly assigned to the experimentally defined number of modules and module size. It is consistent with a widespread influence of MEF2C on the transcriptome as it varies from CS to SD to RS. Furthermore, most of the enriched modules had sleep-condition-related eigengenes indicating that sleep condition was the principal component factor driving their coordinated expression.

We identified two modules (M04 and M10) whose module eigengenes were more positively correlated with CS or ad lib sleep or, conversely, more negatively correlated with SD and RS samples (*Figure 2B* and *Figure 2—figure supplement 1*), indicating that the primary component driving their pattern of coexpression is CS versus SD and RS conditions. These two modules were of interest due to significant enrichment of several major categories of genes (*Figure 2A*): (1) DEGs in neuronal *Mef2c conditional KO* cortex (Mef2c-cKO$^{Emx1}$; *Harrington et al., 2016*), (2) excitatory and inhibitory neuronal genes, (3) genes downregulated in SD and RS, and (4) genes involved in synaptic function, epilepsy and cognitive pathologies including autism (*Figure 2D*). Both of these modules have hub genes encoding receptors, ionic channels or cell adhesion, and identified as autism and epilepsy risk factors (for complete GO of M04 and M10 and other modules, see *Supplementary file 4*). These characteristics are generally consistent with modulation by the MEF2C transcription factor and suggest the modulation by MEF2 is closely associated with sleep conditions. Also, we observe significant enrichment for Mef2c downstream genes (*Harrington et al., 2016*) in 15 out of 59 modules, which is significantly greater than randomly assigning the genes to simulated 59 modules of the same size (Monte Carlo p-value: 0.001, N: 1000 iterations).

## The role of MEF2C in the transcriptomic response to sleep loss

An association of MEF2C target genes was observed by comparing the CS to SD DEGs; we found enrichment for DEGs in Mef2c-cKO$^{Emx1}$ compared to WT cortex involved in regulating synaptic strength (*Figure 1E,G*, *Supplementary file 3*). Additionally, modules of correlated gene expression across sleep conditions showed a similar enrichment of putative MEF2C targets, pattern of expression and GO as the DEGs across sleep conditions (*Figure 2A*, *Supplementary file 5*).

To better understand the role of MEF2C across sleep conditions, we examined a conditional knockout mouse using a Cre-loxP system. We crossed a *Mef2c*$^{f/f}$ mouse line (*Arnold et al., 2007*; *Zang et al., 2013*) with a transgenic *Camk2a-Cre* line and examined these Mef2c-cKO$^{Camk2a-Cre}$ mice (see Materials and methods for details) for their response to sleep loss. *Mef2c*$^{f/f}$ mice (see Materials and methods for comparisons to WT DEGs in response to SD) had 767 DEGs within 12,403 total expressed genes from CS (n=4) to SD (n=3); whereas, strikingly, the Mef2c-cKO$^{Camk2a-Cre}$ mice had only 36 DEGs within 12,620 total expressed genes in CS (n=3) to SD (n=3; *Figure 3A, B, C*). Both the *Mef2c*$^{f/f}$ and the Mef2c-cKO$^{Camk2a-Cre}$ genotypes expressed similar numbers of FC genes as the WT, using a CPM $\geq$1.0, but the responses of Mef2c-cKO$^{Camk2a-Cre}$ to sleep conditions were attenuated to ~7% of the *Mef2c*$^{f/f}$ responses, indicating the essential role of *Mef2c* in the transcriptome response to sleep loss.

Although the observed attenuation of *Mef2c*$^{f/f}$ mouse DEGs from CS to SD induced by the conditional loss of *Mef2c* is striking, it is of note that the number of CS to SD DEGs comparing WT to *Mef2c*$^{f/f}$ also shows some attenuation. We further investigated the differences between WT and *Mef2c*$^{f/f}$ DEGs by down-sampling and permutation of the WT samples so that the numbers of observations between the two cohorts was comparable (CS (n = 4) and SD (n = 3)). We observed increased variability in gene expression and such increased variability can lead to decreased number of observed DEGs (*Supplementary file 9*). However, the decrease due solely to increased variability does not appear of sufficient magnitude to fully account for the difference between WT and *Mef2c*$^{f/f}$ mice.

A potential contributing factor could result from an effect of the LoxP insertions on MEF2C expression. However, an assessment of the specific genomic expression showed only minor variability in exon expression between WT, *Mef2c*$^{f/f}$ and Mef2c-cKO$^{Camk2a-Cre}$ samples with only the expected specific reduction of the floxed exon in Mef2c-cKO$^{Camk2a-Cre}$ mice (*Figure 4*).

Another potential source of difference is from small differences in gene expression due to genetic background. The *Mef2c*$^{f/f}$ line employed in this study has been employed in several other studies of its role in synaptic function (*Barbosa et al., 2008*; *Harrington et al., 2016*; *Rajkovich et al., 2017*), but this does not rule out differences in transcriptional expression due to genetic background. While the *Mef2c*$^{f/f}$ mice were backcrossed to the same genetic background of the WT mice for five generations, there could still be variability in gene expression due to incomplete congenicity and/or

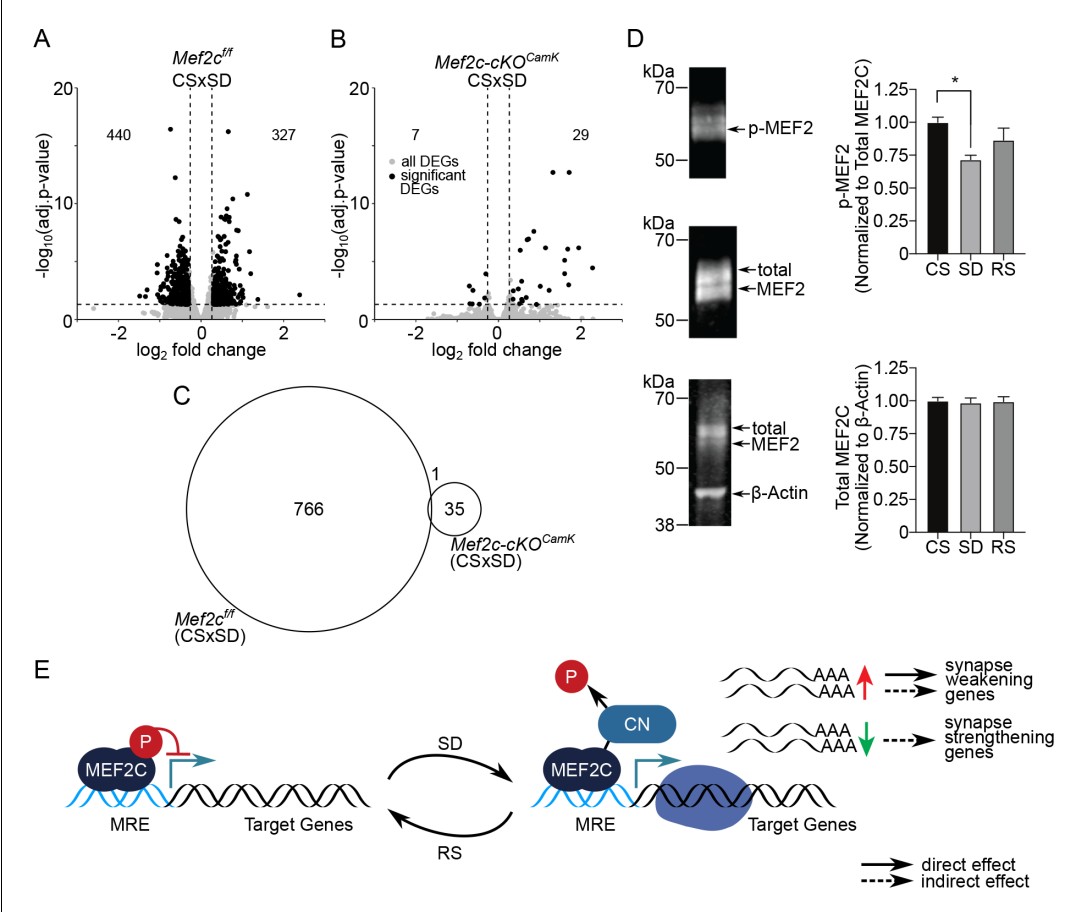

**Figure 3.** MEF2C, its de-phosphorylation and essential role in sleep-loss-mediated differential gene expression. (A) Volcano plot showing the DEGs identified for *Mef2c*^f/f and another, (B) for Mef2c-cKO^Camk2a-Cre between CS to SD sleep condition. DEGs (adj. P-val <= 0.05; absolute log₂(change) => 0.3) are indicated by black dots. (C) A Venn diagram showing overlap between DEGs for *Mef2c*^f/f and Mef2c-cKO^Camk2a-Cre between CS and SD sleep states. (D) Immunoprecipitation of MEF2C to detect MEF2C, Phospho-S396 MEF2C Phospho-S396 and total MEF2C for each sleep condition. Phospho-MEF2 (top blot) re-labeled from immunoprecipitation of total MEF2C (middle blot) was quantified and normalized to total MEF2C signal for each sample (n = 4 samples/condition). Total MEF2C from total cell lysate was quantified and normalized to B-actin (bottom blot). Data reported as mean +/- SEM. Statistical significance was determined by one-way ANOVA with Tukey's multiple comparisons test (interaction = p < 0.05, post-hoc test comparing CS to SD *=p < 0.05). (E) Model showing role of de-phosphorylated, activated MEF2C, increased relative to total MEF2C by loss of sleep and leading to synapse remodeling. MRE = Mef2 response element, CN = calcineurin.

epigenetic variability. Interestingly, an examination of the overlapping DEGs comparing CS to SD from WT and *Mef2c*^f/f mice used in this study (*Supplementary file 10*, Tab 'Overlap'), shows a gene ontology represented by genes controlling synaptic function (*Supplementary file 10*, Tab 'Gene Ontology'), indicating that sleep-condition regulation of synaptic genes is maintained between the two genotypes. Furthermore, the synaptic and EEG phenotypes of these genotypes across sleep conditions are indistinguishable as shown below (*Figure 6—figure supplement 1* and *Figure 5*).

MEF2 transcriptional activity is increased upon activity-dependent dephosphorylation of S396 (*Flavell et al., 2006*). Expression of a constitutively-active form of MEFC (MEF2C-VP16) is sufficient to reduce cortical excitatory synapses (*Harrington et al., 2016*), mimicking reduction of excitatory synaptic strength previously reported in response to recovery sleep (*Liu et al., 2010*). This led to our speculation that sleep loss might decrease MEF2C phosphorylation at S396. Using a phosphorylation site-specific antibody (*Flavell et al., 2006*), we examined the MEF2C phosphorylation status in FC under three sleep conditions (i.e. CS, SD and RS). We found that compared to CS, SD produced a significant reduction of MEF2C P-S396 (*Figure 3D*), consistent with an increase in MEF2C activity during SD, but without altering total MEF2C protein levels (*Figure 3D*). In contrast, RS did not significantly alter MEF2C P-S396 levels (*Figure 3D*), suggesting that SD triggers signaling events to

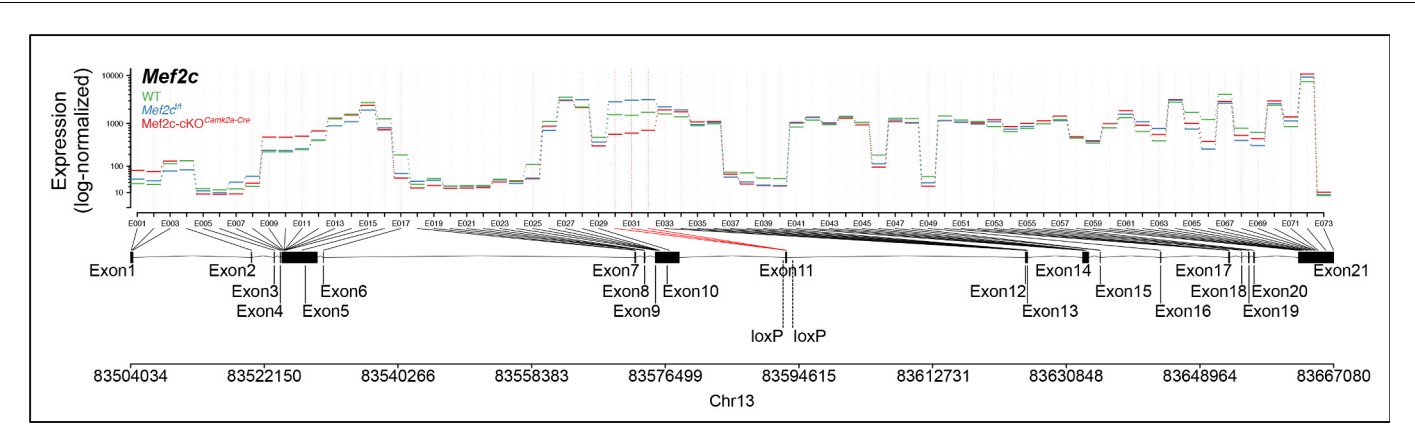

**Figure 4.** Differential exon usage analysis showing minor variability in exon expression between WT, *Mef2c*[f/f] and Mef2c-cKO[Camk2a-Cre] samples, also indicating specific reduction of floxed exon in Mef2c-cKO[Camk2a-Cre].

transiently activate MEF2C-dependent transcription and promote the weakening and/or elimination of excitatory synapses.

As noted above (*Figure 1E*), CS to SD and RS downregulated DEGs were enriched for genes that were downregulated in an earlier study comparing WT to Mef2c-cKO[Emx1] in control conditions (*Harrington et al., 2016*). Similarly, the upregulated genes from CS to SD and RS are enriched for

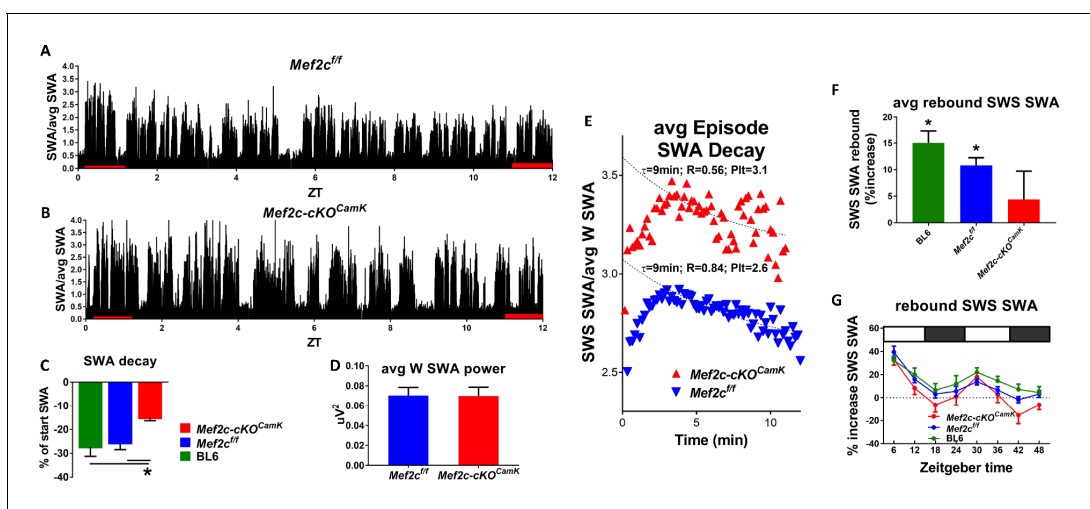

**Figure 5.** Loss of Mef2c increases sleep need and decreases its resolution. (A) SWA power (normalized to average SWA power over 24 hr) declines over the light phase as shown in a Mef2c[f/f] mouse. (B) The decline is less apparent in an example from a Mef2c-cKO[Camk2a-Cre] mouse. (C) Pooled data (Mef2c[f/f], n = 6; Mef2c-cKO[Camk2a-Cre], n = 11; C57BL/6, n = 6) SWA power decay over the light phase for WT (BL/6), Mef2c[f/f] and Mef2c-cKO[Camk2a-Cre]. (D) The average waking SWA power in 24 hr is similar for Mef2c-cKO[Camk2a-Cre] and Mef2c[f/f] and is used as a normalizing factor to assess averaged SWS episode power. (E) SWS-SWA power over time from an average of SWS episodes aligned from the time of transition from waking to SWS, set to t = 0. The time constant of decay, $\tau$ = 8.4 min, was calculated by best fit of the decay of power for the Mef2c[f/f] but the decay fit was indeterminate for Mef2c-cKO[Camk2a-Cre] (see Materials and methods section for details). By setting $\tau$ = 8.4 min, a best fit to single exponential decay provided a plateau value (minimal value), an extrapolated peak value and an indication of goodness of fit in comparison to Mef2c[f/f]. The SWS-SWA power for Mef2c-cKO[Camk2a-Cre] mice is greater than Mef2c[f/f] as determined by the plateau value (Plt) and the peak value. The quality of the fit is reduced as indicated by the $R^2$ value. (F) The average SWS-SWA power of each genotype following repeated, regular bouts of 4 hr SD, was significantly increased compared to SWS-SWA power under baseline conditions. Mef2c-cKO[Camk2a-Cre] mice failed to show this rebound response to sleep loss. (G) The time course of rebound across the 8 × 4 hr SD periods indicates that the lack of rebound in Mef2c-cKO[Camk2a-Cre] is particularly prominent during the dark phase.
The online version of this article includes the following figure supplement(s) for figure 5:

**Figure supplement 1.** Loss of Mef2c does not influence sleep/waking time, but modestly alters spectral power distribution across states.

up-regulated WT to Mef2c-cKO$^{Camk2a-Cre}$ genes. In other words, the DEGs of Mef2c-cKO$^{Camk2a-Cre}$ (compared to WT) in control conditions are similar to the DEGs of CS to SD in WT mice. However, in the Mef2c-cKO$^{Camk2a-Cre}$ used in our study, there is little response to SD (*Figure 3B,C*), rather the DEGs of the WT sleep loss response are now differentially expressed in CS conditions in Mef2c-cKO-$^{Camk2a-Cre}$ mice, potentially, a compensatory response, no longer responsive to sleep loss.

## The role of *Mef2c* in the synaptic response to sleep loss

Based on our transcriptome analyses, SD and RS induced many DEGs affecting synaptic strength and synaptic cellular components that required the MEF2C transcription factor (*Figures 1*, *2* and *3*). MEF2C can regulate forebrain excitatory synapse strength of pyramidal neurons by decreasing synaptic density (*Harrington et al., 2016*) and frequency (*Flavell et al., 2006*). Down-scaling of cortical synaptic strength is associated with sleep (*de Vivo et al., 2017*; *Diering et al., 2017*). Previous observations indicate that acute loss of sleep is associated with increased miniature excitatory post-synaptic (mEPSC) frequency and amplitude and that recovery sleep reverses these changes in cortical neurons recorded from C57BL/6 mouse FC (*Liu et al., 2010*). However, neither the involvement of changes in probability of release nor changes in active excitatory synapse number and the role of *Mef2c* have been previously considered.

To directly examine sleep-associated changes in synaptic strength, the same conditions as employed for the transcription analyses (i.e. CS, SD, RS) were applied to *Mef2c*$^{f/f}$ male mice. Acute ex vivo slices of FC were prepared from these mice at ZT = 6 hr, for whole cell voltage-clamp recordings from layer 2–3 pyramidal neurons. Because sleep-loss is associated with an increased extracellular concentration of adenosine throughout the forebrain (*Porkka-Heiskanen et al., 2000*; *Porkka-Heiskanen et al., 1997*) which decreases glutamatergic mEPSC frequency and amplitude (*Brambilla et al., 2005*; *Scanziani et al., 1992*), all recordings were made in the presence of adenosine receptor 1 and 2 blockade (cyclopentyl-theophyline, 1 μM).

As expected, we observed an increase of mEPSC frequency and amplitude after SD and the recovery to CS-levels in RS condition in the WT and *Mef2c*$^{f/f}$ mice (*Figure 6A–F*; *Figure 6—figure supplement 1*; *Supplementary file 6*).

To determine if the SD-induced increase in mEPSC frequency was caused by an increased presynaptic release probability or an increase in synapse number, we examined the paired-pulse ($P_2/P_1$) ratio from CS to SD and RS (*Figure 7A,B*).

The $P_2/P_1$ ratio tended toward an increase from CS to SD condition (this is consistent with a decreased probability of release), suggesting that the SD-associated increase in mEPSC frequency resulted from an increased number of active synapses. However, for CS to RS, the observed significant increase in $P_2$ to $P_1$ ratio and the normalization of mEPSC frequency after RS is due to either a decrease in release probability, a decrease in active synapse number or both.

In Mef2c-cKO$^{Camk2a-Cre}$ mice, mEPSC frequency and amplitude was greater than in *Mef2c*$^{f/f}$ in CS condition. Consistent with the lack of change in the Mef2c-cKO$^{Camk2a-Cre}$ transcriptome, a significant change in frequency and mEPSC amplitude in response to SD and RS sleep conditions was absent (*Figure 6G–L*, *Supplementary file 7*). Finally, as expected from the absence of SD or RS induced effects on mEPSCs, the Mef2c-cKO$^{Camk2a-Cre}$ mice had no change in $P_2/P_1$ ratio across sleep conditions (*Figure 7C,D*). These observations support a necessary role for *Mef2c* in SD and RS associated changes in synaptic strength for FC layer 2/3 excitatory neurotransmission.

## The role of *Mef2c* in the generation and resolution of sleep need

Sleep need, inferred from the duration of time in waking without sleep (*Borbély, 1982*; *Franken et al., 2001*), is directly correlated with the ensuing SWS-SWA power calculated from surface EEG recorded from control *Mef2c*$^{f/f}$ and Mef2c-cKO$^{Camk2a-Cre}$ mice. In WT and *Mef2c*$^{f/f}$ mice, SWS-SWA is greatest at the start of the sleep phase and resolves over the sleep phase course as the mice spend time in SWS (ZT 0–12 hr; *Figure 5A,C,E*). In contrast, Mef2c-cKO$^{Camk2a-Cre}$ mice express, for their averaged SWS-SWA episodes, higher SWA (*Figure 5E*) that fails to normally resolve over the course of the sleep phase (*Figure 5B,C*). Another indicator of increased sleep need is increased consolidation of SWS as suggested by a rightward shift in the cumulative histogram of SWS episode duration going from CS to SD conditions (*Bjorness et al., 2016*), which is occluded in Mef2c-cKO-$^{Camk2a-Cre}$ mice (*Figure 5—figure supplement 1A*).

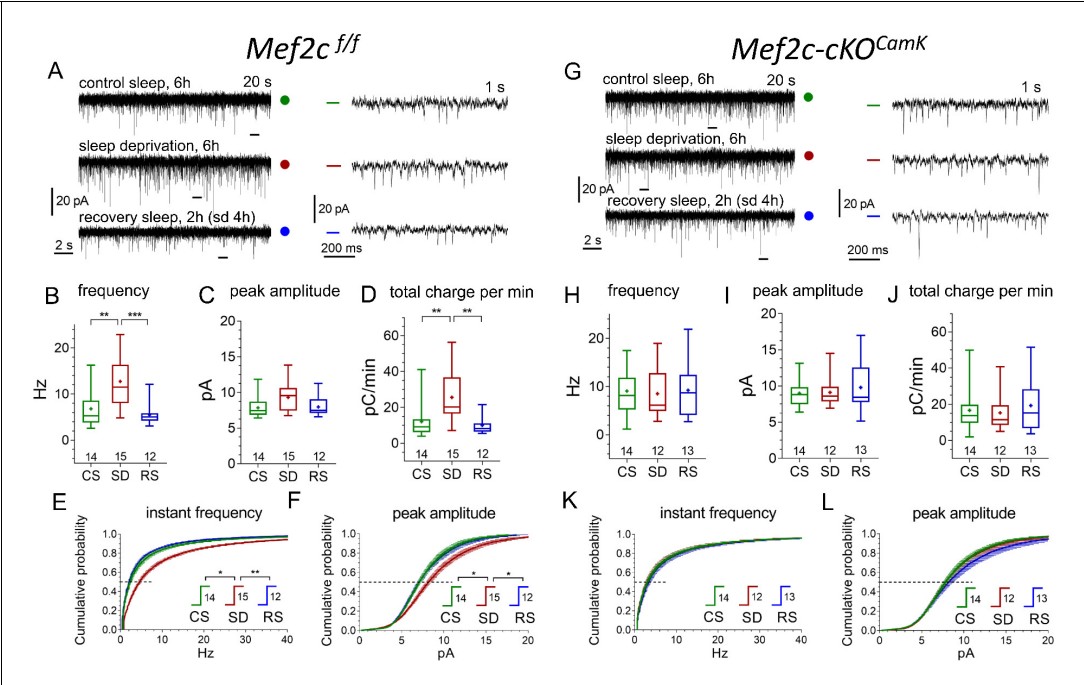

**Figure 6.** Conditional Mef2c knock-out in forebrain excitatory neurons eliminates sleep/wake remodeling of synaptic excitatory inputs to pyramidal neurons in anterior cingulate cortex slices. (A, G) Representative recordings of miniature excitatory postsynaptic current (mEPSC) traces (20 s, left panels, and expanded 1 s traces on the right corresponding to time bars underneath original traces) obtained for three different experimental sleep/wake conditions, CS (green), SD (red), and RS (blue), for *Mef2c*f/f mice (A) and Mef2c-cKO$^{Camk2a-Cre}$ mice (G). (B,C,D) and (H,I,J) illustrate mEPSC functional parameters obtained from *Mef2c*f/f and Mef2c-cKO$^{Camk2a-Cre}$ (number of cells for each condition shown above X axis condition label; also see *Figure 6—figure supplement 1* for comparison to WT). Averaged cumulative probability histograms for instant frequency and peak amplitude show increased frequency and amplitude with SD compared to CS or RS for *Mef2c*f/f (E, F) but not for Mef2c-cKO$^{Camk2a-Cre}$ mice (K, L). See Materials and methods and *Supplementary file 6* for details and statistics.

The online version of this article includes the following figure supplement(s) for figure 6:

**Figure supplement 1.** Similar responses of WT and *Mef2c*f/f mEPSCs to SD and RS conditions.

With respect to arousal-state duration (waking vs SWS vs REM), Mef2c-cKO$^{Camk2a-Cre}$ mice had no apparent significant differences and there were minimal differences in the power distribution across the frequency spectrum (*Figure 5—figure supplement 1B,C*). In response to SD, the Mef2c-cKO-$^{Camk2a-Cre}$ mice show an attenuated or absent rebound of SWS-SWA (*Figure 5F,G*). These observations suggest that a loss of MEF2C function both increases the need for sleep and attenuates its resolution during SWS.

Since MEF2C Haploinsufficiency Syndrome (MCHS) in humans is associated with sleep-disruption (*Harrington et al., 2020*; *Paciorkowski et al., 2013*), we examined constitutive gene dosage on SWS-SWA power of a global *Mef2c* heterozygous gene-deletion mouse model (*Mef2c*+/-) of MCHS under CS and SD conditions (*Figure 8*).

Under CS conditions, SWS-SWA power resolved across the sleep phase, unlike the Mef2c-cKO-$^{Camk2a-Cre}$ mice, indicative of a resolution of sleep need. Similarly, *Mef2c*+/- mice showed resolution of sleep need over an average SWS episode, again, unlike Mef2c-cKO$^{Camk2a-Cre}$ mice; however, the rate of resolution (as indicated by the rate of decay of SWA in an average SWS episode) was slowed. On the other hand, SD did not induce the expected rebound SWS-SWA in *Mef2c*+/- mice similar to Mef2c-cKO$^{Camk2a-Cre}$ mice. Another similarity of *Mef2c*+/- mice with the Mef2c-cKO$^{Camk2a-Cre}$ mice is the apparent occlusion of the rightward shift in the cumulative histogram of SWS episode duration going from CS to SD conditions (*Figure 8—figure supplement 1*), associated with increased sleep need in WT mice (*Bjorness et al., 2016*). The *Mef2c*+/- mice are already shifted to the right compared to *Mef2c*+/+ mice in both CS and SD conditions. Furthermore, the *Mef2c*+/- mice had an abnormal relative increase in waking SWA power (in the high delta/low theta range of 4–5 Hz; *Figure 8—figure supplement 1*).

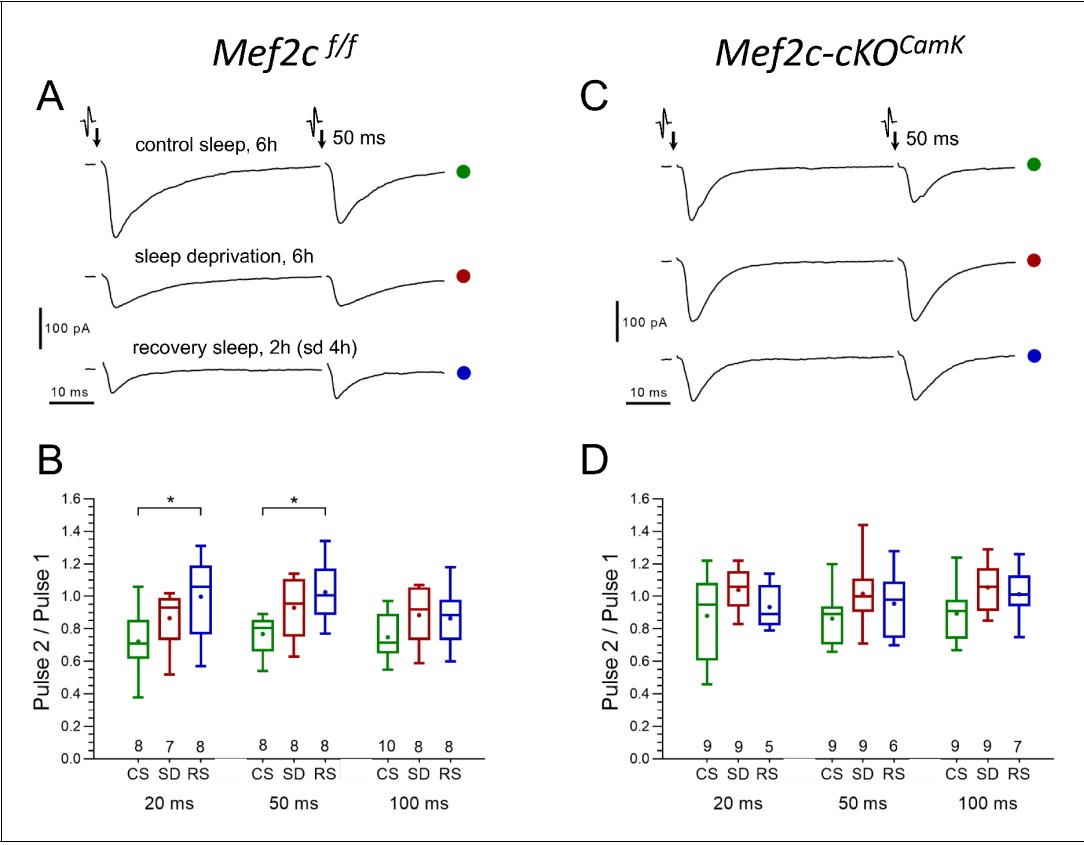

**Figure 7.** No significant differences in presynaptic glutamate release probability between CS and SD are found in either control or Mef2c-cKO$^{Camk2a-Cre}$ mice. (**A and C**) Representative paired-pulse recordings of excitatory postsynaptic currents (evoked EPSC) evoked in layer 2/3 pyramidal neurons by two brief electrical stimulations of axon terminals in layer I, 50 ms interval, obtained for three different experimental sleep/wake conditions (CS, SD, RS) imposed on *Mef2c*$^{f/f}$ mice (**A**) and Mef2c-cKO$^{Camk2a-Cre}$ mice (**C**), respectively. The probability of presynaptic release (P$_{release}$), is inversely proportional to the ratio of EPSC amplitudes (1/P$_{release}$ ~ P$_2$/P$_1$); The P$_2$/P$_1$ ratio, obtained for three different experimental sleep/wake conditions (CS, SD, RS) and calculated separately for three different inter-pulse intervals (20, 50 and 100 ms), in *Mef2c*$^{f/f}$ (**B**) vs. Mef2c-cKO$^{Camk2a-Cre}$ mice (**D**). The plots (see Materials and methods, numbers of experiments for each sleep/wake conditions shown underneath boxes), show no significant differences between CS and SD conditions in either control or in Mef2c-cKO$^{Camk2a-Cre}$ mice.

By comparing differential gene expression under CS conditions in each of the *Mef2c* models, we found a significant overlap of genes changing in the *Mef2c*$^{+/-}$ and Mef2c-cKO$^{Camk2a-Cre}$ compared to their respective controls (142 overlapping genes; p-value=5.28e-08, Fisher's exact test (*Figure 8—figure supplement 2*, *Supplementary file 8*)). The identified overlapping gene set can provide clues into the molecular mechanisms underlying a requisite *Mef2c* sufficiency. Moreover, the non-overlapping genes can reflect either developmental effects or non-neuronal effects present in the haploinsufficiency model, but absent in the Mef2c-cKO$^{Camk2a-Cre}$ mice. Taken together, these findings suggest constitutive loss of one *Mef2c* allele is sufficient to alter the normal response to sleep loss, but the loss appears to be at least partially compensated when spontaneous sleep remains undisturbed. More acute, complete loss of *Mef2c* function as with the Mef2c-cKO$^{Camk2a-Cre}$ mouse, even further disrupts the homeostatic sleep loss response.

## Discussion

Sleep-loss-induced genomic alterations are remarkably extensive, involving almost half of the expressed genome of the frontal cortex. About two thirds of the significant changes in gene expression observed in response to sleep deprivation were maintained when 2 hr of recovery sleep followed 4 hr of sleep loss indicating that direct recovery of sleep deprivation-induced transcriptional changes is delayed for at least the 2-hr observation time employed in this study. This result suggests

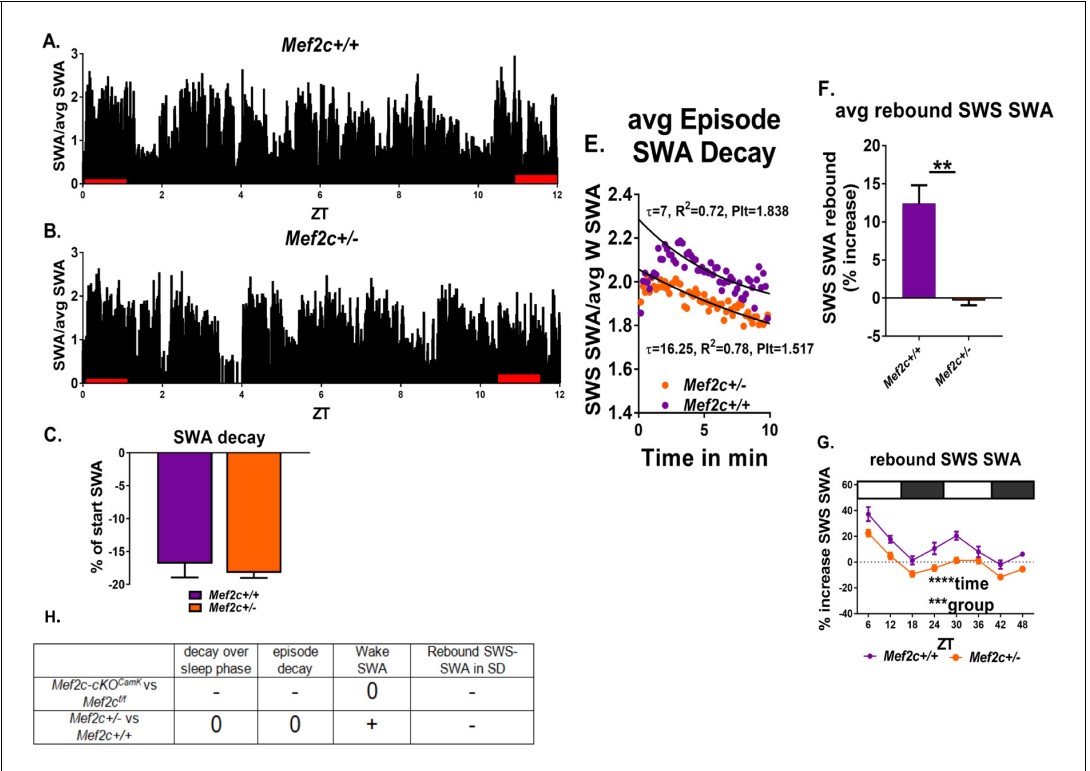

**Figure 8.** *Mef2c^{+/-}* mice resolve sleep need in undisturbed conditions but lack a rebound increase of SWS-SWA power in response to sleep loss. (A,B) Average SWA power per 10 s epoch decreases during the sleep phase (ZT = 0 to 12 hr) to the same extent for *Mef2c^{+/+}* (n = 6) and *Mef2c^{+/-}* (n = 6) mice (C). (D) Twenty-four hour average waking SWA power is greater for *Mef2c^{+/-}* mice. (E) Decay of SWA power (normalized to the average 24 hr waking SWA) during an average SWS episode is preserved in the *Mef2c^{+/-}* mice but with an increased τ, associated with increased sleep need. (F, G) Rebound increased SWS-SWA in response to SD is absent in *Mef2c^{+/-}* mice. The SD was administered in repeated 6 hr cycles of 4 hr SD and 2 hr RS. (H) A table showing responses of Mef2c-cKO^{Camk2a-Cre} and *Mef2c^{+/-}* compared to their respective controls (see *Figure 5*).

The online version of this article includes the following figure supplement(s) for figure 8:

**Figure supplement 1.** *Mef2c^{+/-}* mice are similar to *Mef2c^{+/+}* mice with respect to SD evoked increase in SWS episode consolidation.

**Figure supplement 2.** Total of 2565 differentially expressed genes across *Mef2c^{f/f}* and Mef2c-cKO^{Camk2a-Cre} under CS conditions.

additional post-transcriptional mechanisms are also key to sleep-loss molecular mechanisms as for example, in synaptosomes, the translation of DEGs to proteins is at least, in part, dependent on sleep (*Noya et al., 2019*) or, more generally, the sleep loss associated increase in phosphorylation of the proteome (*Wang et al., 2018*).

The transcriptional changes shared between CS to SD and CS to RS, included increased expression of genes involved in protein anabolism (including genes encoding ribosomal components and enzymes controlling their activity), protein catabolism and mobilization of energy resources (including genes encoding mitochondrial structural components and enzymes). These altered expression patterns are associated especially with non-neuronal brain tissue. In contrast, there is enrichment for decreased expression of genes primarily linked to neuronal cells, for the control of synaptic strength. These sleep condition associations relate protein and energy metabolism to neuronal synaptic strength in much the same way as the energy-related metabolite, adenosine, mediates a synaptic homeostasis of glutamate synaptic activity (*Brambilla et al., 2005*), which is part of a neuronal-glial circuit limiting glutamatergic activity. This circuit is an essential part of the CNS response to sleep loss (*Bjorness et al., 2016*; *Bjorness et al., 2009*; *Brambilla et al., 2005*; *Greene et al., 2017*). Because we conditionally deleted Mef2c in neurons induced by Camk2a-Cre expression and did not see large transcriptomic changes in neurons or non-neurons in response to SD, we can speculate that the observed changes in non-neuronal genes during SD were likely secondary to loss of Mef2c function that was restricted to neurons. A contributing factor to SD evoked secondary transcriptional changes in non-neuronal cells could result from Mef2c-dependent changes in SD evoked increases in

excitatory synaptic strength that we observed along with its absence after neuronal loss of Mef2c. Adenosine, released in response to increased glutamatergic synaptic activity associated with SD and acting to decrease this same activity (*Brambilla et al., 2005*), affects non-neuronal cells through non-neuronal adenosine receptors and/or by affecting glial adenosine kinase (*Bjorness et al., 2016*; *Greene et al., 2017*). Accordingly, adenosine can be a molecular mediator (although not necessarily the only mediator) for these Mef2c-dependent secondary effects on the non-neuronal transcriptome.

We would also like to note that the design of our current study only examined gene expression changes after 2 hr of recovery sleep. It is possible that many genes require greater than 2 hr of recovery sleep before returning to baseline. Future studies that use additional longer time periods of recovery sleep can examine this possibility.

Coordination of the extensive change in transcriptome in response to loss of sleep requires the function of the transcription factor MEF2C, and sleep loss appears to increase MEF2C activity via the well-studied, calcium-dependent dephosphorylation of MEF2C in frontal cortex. The MEF2C-dependent, sleep loss, DEGs encode proteins modulating cellular and systemic functions. These same functions may also be expected to be affected by sleep loss in an MEF2C-depedent manner, and indeed, the normal reduction of synaptic strength and activity associated with recovery sleep (RS) is absent in the Mef2c-cKO$^{Camk2a-Cre}$ mice. Notably, in WT, there is an acute increase in synaptic activity and strength induced by prolonged waking (SD) that we observe in association with a down-regulation of genes controlling synaptic strength (*Figure 1*, *Figure 6*). The SD, down-regulation of synaptic DEGs is presumably not translated during the prolonged waking time, but with recovery sleep, the effects become manifest. Nonetheless, the normal increase in synaptic activity resulting from prolonged waking is also absent in Mef2c-cKO$^{Camk2a-Cre}$ mice, indicating an abnormal synaptic response both to the acute loss of sleep and to recovery sleep in the absence of MEF2C function.

In a normal waking state, MEF2C may function mainly as a transcriptional repressor when phosphorylated at S396 (*Harrington et al., 2016*). With loss of *Mef2c*, MEF2C will no longer be available for activation of specific target genes resulting in both de-repression from decreased phosphorylated MEF2C and loss of facilitation of transcription by dephosphorylation of S396 of MEF2C. Many of the DEGs comparing CS to SD conditions in WT animals showed significant overlap with genes dysregulated in Mef2c-cKO$^{Emx1}$ mice. These same genes no longer show differential expression comparing CS to SD in Mef2c-cKO$^{Camk2a-Cre}$ mice in the present study, implicating an essential role in WT mice for MEF2C in sleep-condition-dependent transcriptome change. In Mef2c-cKO$^{Emx1}$ mice, similar widespread, abnormal gene expression is apparent, even in the absence of experimentally induced sleep loss. A portion of such widespread gene expression alterations may be in response or compensation to the more chronic absence of MEF2C-mediated response to sleep loss, since recombination of the floxed-alleles occurred early in embryonic brain development and may alter cortical circuit development.

Our findings show that loss or reduction of MEF2C function alters the response to sleep loss. Postnatal loss of MEF2C in forebrain excitatory neurons results in an enhanced SWS-SWA rebound and an attenuated resolution of the rebound. To the extent that SWS-SWA reflects sleep need, loss of MEF2C function increases sleep need in response to a loss of sleep compared to wild type MEF2C response, and it attenuates the sleep-mediated resolution of sleep need. We would like to point out that our comparisons of Mef2c-cKO$^{Camk2a-Cre}$ mice are always to mice with the floxed allele, *Mef2c*$^{f/f}$. However, similar comparisons to mice that only express Cre would also be another valid and important comparison.

Taken together, these findings suggest a model of the sleep loss response that reflects a MEF2C-dependent, metabolically driven non-neuron to neuron homeostatic sleep circuit. This circuit can act to promote protein and energy metabolism and constrain synaptic activity in response to sleep loss.

## Materials and methods

**Key resources table**

| Reagent type (species) or resource | Designation | Source or reference | Identifiers | Additional information |
| --- | --- | --- | --- | --- |

*Continued on next page*

*Continued*

| Reagent type (species) or resource | Designation | Source or reference | Identifiers | Additional information |
|---|---|---|---|---|
| Genetic reagent (*Mus. Musculus*) | C57BL/6 background | Jackson Laboratories | | WT (8–19 weeks old) |
| Genetic reagent (*Mus. Musculus*) | *Mef2c*$^{f/f}$ (C57BL/6J background) | *Arnold et al., 2007*; PMID:17336904 | RRID:MGI:3719006 | *Mef2c*$^{f/f}$ (8–19 weeks old): knock-in of loxP sites flanking exon 11 of the *Mef2c* gene, a control for the Mef2c-cKO$^{Camk2a-Cre}$ |
| Genetic reagent (*Mus. Musculus*) | C57BL/6 background, B6.Cg- Tg(*Camk2a-Cre*) T29-1Stl/J | Jackson Laboratories | | Mef2c-cKO$^{Camk2a-Cre}$ (8–19 weeks old): CRE-mediated recombination of the *Mef2c*$^{f/f}$ loxP sites, resulting in a region and cell type specific conditional knockout Mef2c-cKO$^{Camk2a-Cre}$ |
| Genetic reagent (*Mus. Musculus*) | *Mef2c*$^{+/-}$ (backcrossed to C57BL/6J) | *Harrington et al., 2020*. PMID:32418612. | | |
| Antibody | Anti-MEF2C (Mouse, Monoclonal) | Novus | Cat. # NBP2-00493 | Immunoprecipitation (2 ul (ug) per sample) |
| Antibody | anti-MEF2C (Rabbit, Monoclonal) | Abcam | Cat. #: ab197070; RRID:AB_2629454 | (1:2500) (WB) |
| Antibody | anti-phospho-MEF2 (Rabbit, Polyclonal) | Phospho-peptide affinity purified in-house (as described in *Flavell et al., 2006*. PMID:16484497) | | (1:200) (WB) |
| Antibody | Anti-beta-actin (Rabbit, Monoclonal) | Cell Signaling | Cat. #: 4970S RRID:AB_2223172 | (1:1000) (WB) |
| Antibody | IRDye 800CW Goat anti-Rabbit IgG Secondary Antibody | Li-Cor | Cat. #: 926–32211 | (1:20,000) (WB after IP) |
| Sequence-based reagent | Mef2C KO F1 | This paper (GGGAACCTGACAAATGTGGG) | PCR Primers | Products of 2 kb for non-recombined *Mef2c*$^{f/f}$ and 1 kb for recombined Mef2c-cKO$^{Camk2a-Cre}$ alleles |
| Sequence-based reagent | Mef2C KO R2 | This paper (GTGCATGGCACAGACTACTAGC) | PCR Primers | |
| Commercial assay or kit | RNeasy mini kit | Qiagen | Cat. #. 74104 | RNA purification |
| Commercial assay or kit | TrueSeq Stranded mRNA Library Prep | Illumina | Cat. #. 20020596 | RNA-seq library preparation |
| Commercial assay or kit | Agilent Bioanalyzer High-Sensitivity DNA chip | Agilent | Cat. #. 5067–1504 | |
| Commercial assay or kit | DC Protein Assay Kit II | Bio-Rad | Cat. #: 5000112 | |
| Commercial assay or kit | 7.5% Mini-PROTEAN TGX Stain-Free Protein Gels, 15 well, 15 µl | Bio-Rad | Cat.#: 4568026 | WB after IP |
| Commercial assay or kit | Trans-Blot Turbo Midi PVDF Transfer Packs | Bio-Rad | Cat. #: 1704157 | WB after IP |
| Commercial assay or kit | Odyssey blocking buffer | Li-Cor | Cat. #: NC9877369 | WB after IP |
| Chemical compound, drug | TRIzol reagent | Thermo Fisher Scientific | Cat. #. 15596026 | RNA extraction |
| Software, algorithm | GraphPad Prism | GraphPad Prism https://graphpad.com | | |
| Software, algorithm | FASTQC | Babraham Bioinformatics https://www.bioinformatics.babraham.ac.uk/projects/fastqc | | |
| Software, algorithm | Trimmomatic | *Bolger et al., 2014* | | |

*Continued on next page*

*Continued*

| Reagent type (species) or resource | Designation | Source or reference | Identifiers | Additional information |
|---|---|---|---|---|
| Software, algorithm | STAR | *Dobin et al., 2013* | | |
| Software, algorithm | R | R Foundation https://www.r-project.org | | |
| Software, algorithm | HTSeq | *Anders et al., 2015* | | R package |
| Software, algorithm | edgeR | *McCarthy et al., 2012* *Robinson et al., 2010* | | R package |
| Software, algorithm | DESeq2 | *Anders and Huber, 2010* *Love et al., 2014* | | R package |
| Software, algorithm | UpSetR | *Conway et al., 2017* | | R package |
| Software, algorithm | WGCNA | Weighted gene co-expression network analysis *Langfelder and Horvath, 2008* | | R package |
| Software, algorithm | DEXseq | *Anders et al., 2012* | | R package |
| Software, algorithm | ToppGene | *Chen et al., 2009* https://toppgene.cchmc.org | | Web-tool |
| Software, algorithm | REVIGO | http://revigo.irb.hr *Supek et al., 2011* | | Web-tool |
| Other | Single-cell RNA-seq Dataset | Mouse visual cortex single cell RNA-seq dataset *Hrvatin et al., 2018* | NCBI GEO: GSE102827 | |
| Other | Protein G Plus/A agarose beads | Millpore Sigma | IP05 | Immunoprecipitation (75 ul per sample) |

## Animals

All mice (*Mus musculus*) had a C57BL/6J background, were male and of 8–19 weeks of age at time of data collection. Mice were on a 12 hr light-dark cycle with access to food and water ad libitum. For RNA-seq experiments, we employed three genotypes: wild type (WT); a knock-in of loxP sites flanking exon 11 of the *Mef2c* gene, the second coding exon for the DNA-binding site (*Arnold et al., 2007*)($Mef2c^{f/f}$), a control for the Mef2c-cKO$^{Camk2a-Cre}$; and a CamKII-promoter (*Tsien et al., 1996*), CRE-mediated recombination of the $Mef2c^{f/f}$ loxP sites (founders obtained from Jackson Labs Jackson labs; B6.Cg-Tg(*Camk2a-Cre*)T29-1Stl/J), resulting in a region and cell-type-specific conditional knockout (Mef2c-cKO$^{Camk2a-Cre}$ [*Arnold et al., 2007*]). Primers and primer products are shown along with a map of the *Mef2c* gene (*Figure 9*). Initially, we used a primer pair, Mef2C.2 and Mef2C.3 that gave a product of 650 BP for WT, 750 BP for $Mef2c^{f/f}$ and no product for a CRE-mediated recombined allele to distinguish homozygous and heterozygous floxed knockins from WT mice.

In order to confirm CRE-mediated recombination in the brain, we used another primer pair, Mef2C KO F1 and Mef2C KO R2 (*Figure 9*) that gave products of 2250 BP for non-recombined $Mef2c^{f/f}$ and 1000 BP for recombined Mef2c-cKO$^{Camk2a-Cre}$ alleles. All brain samples of Mef2c-cKO-$^{Camk2a-Cre}$ mice expressed both products as expected.

We also determined that some of the offspring of male $Mef2c^{f/wt}$;*Camk2a-Cre* mice were heterozygote for a single recombined allele, showing $Mef2c^{f/-}$ in tail samples, consistent with CRE-mediated recombination in spermatocytes (*Luo et al., 2020*). Thus, gene-dosage-dependent developmental effects of *Mef2c* loss of function in all cell types may contribute to the transcriptome of Mef2c-cKO$^{Camk2a-Cre}$ mice in addition to the homozygous conditional loss of *Mef2c* function in the brain. Nevertheless, a heat map of differentially expressed genes (DEGs) from Mef2c-cKO$^{Camk2a-Cre}$ mice showed similar clustering of DEGs from samples in CS conditions with, and without (second column *Figure 10*) recombination in the tail.

For in vivo electrophysiology (EEG/EMG analysis for sleep/waking state determination), in addition to the three strains described above, we also included constitutive heterozygous Mef2c

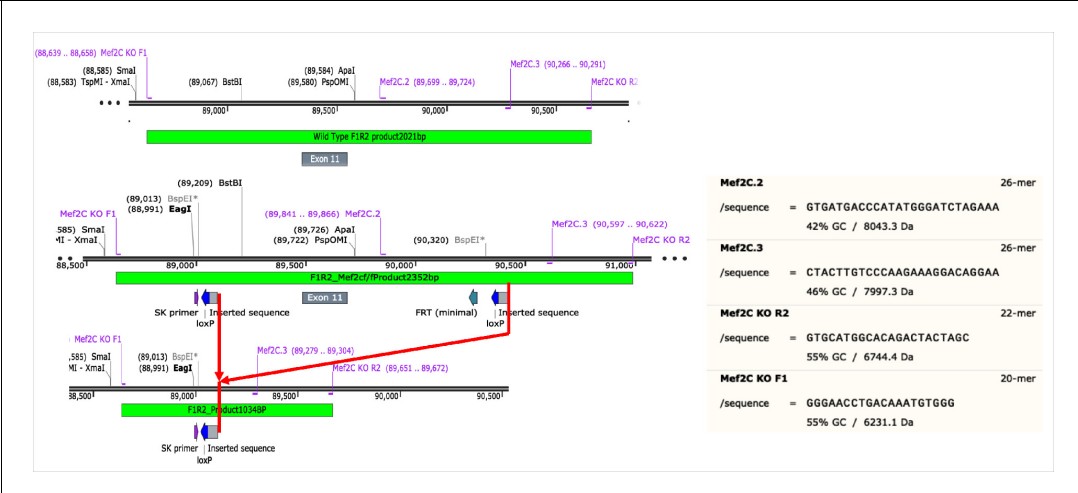

**Figure 9.** Schematic of Mef2c genomic DNA sequence to illustrate locations of knockin sequences of loxP and their recombination product used in this study.

knockouts (*Mef2c^{+/-}*) and their littermate controls (*Mef2c^{+/+}*); generation of these strains has been described previously (*Harrington et al., 2020*).

All experimental procedures were approved by the University of Texas Southwestern IACUC or VA North Texas Health Care System IACUC.

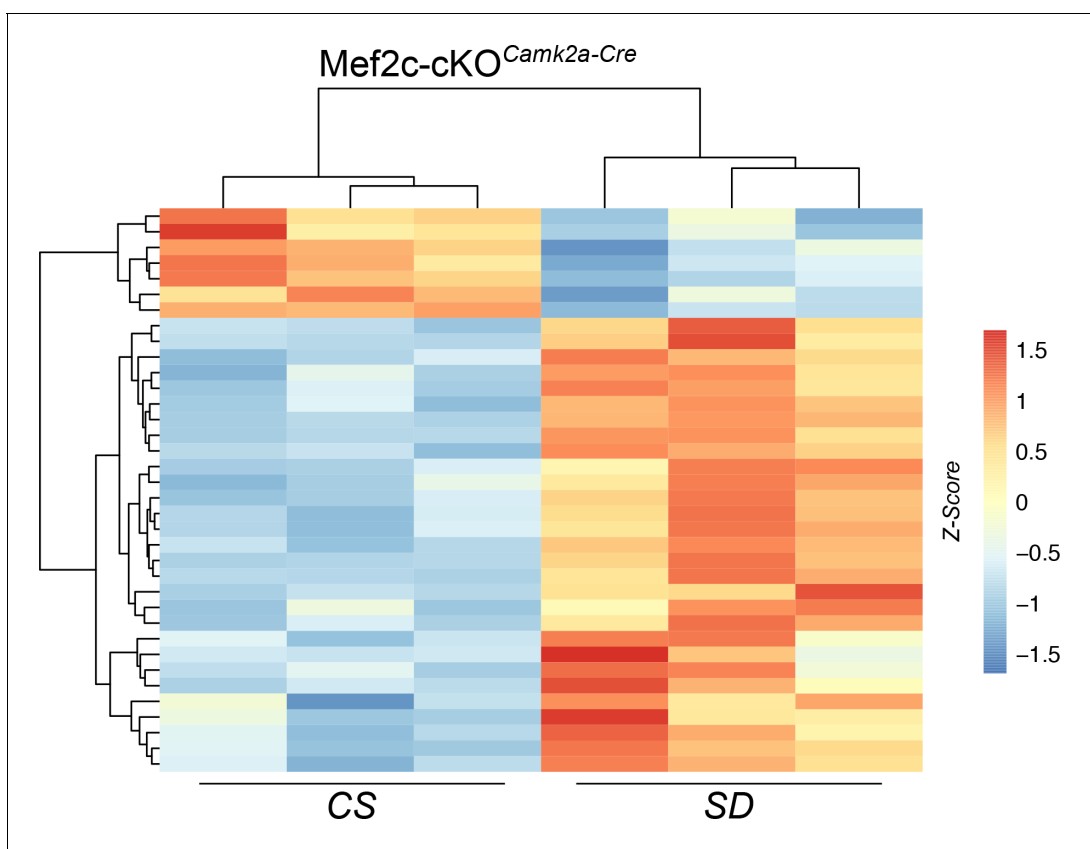

**Figure 10.** Heat map of DEGs from *Mef2c-cKO^{Camk2a-Cre}* mouse samples under CS or SD conditions. The sample from the second column (CS) showed a similar DEG clustering pattern to the other two samples that had germline recombination.

## Sleep deprivation protocol

All mice underwent an accommodation protocol of 3 weeks duration, in 12/12 light/dark schedule, in plexiglass, false-bottomed cages, suspended over a treadmill (see *Video 1*) with ad libitum water and mouse chow as previously described (*Bjorness et al., 2016*). The treadmill (speed of treadmill = 3 cm/sec) was turned on daily for 10 min to accommodate the mice to the slow movement (mice are able to eat and drink with it running) and to clean it. For RNA-seq, 36 hr prior to the experimental day (CS, SD or RS) a dark/dark schedule was instituted. The use of the treadmill prevented transition to sleep, did not increase motor activity (due to the slow speed) and provided a mild, familiar, non-varying, arousing stimuli that reduced the experimental variability associated with more traditional 'gentle handling' techniques. Animals were randomly assigned to CS, SD or RS groups.

## RNA collection, RNA-seq library preparation and Sequencing

Total RNA was extracted from all the samples using TRIzol reagent (Thermo Fisher Scientific, Cat. No. 15596026) and was purified with RNeasy mini kit (Qiagen, Cat No. 74104) according to manufacturer's instructions. All the samples were then randomized in order to reduce the batch effect and submitted to McDermott Sequencing Core at University of Texas Southwestern Medical Center for library preparation and sequencing. Libraries were prepared using TrueSeq Stranded mRNA Library Prep (Illumina, Cat. No. 20020596) as per manufacturer's instructions by McDermott Sequencing Core. The quality and concentration of the libraries was checked on Agilent Bioanalyzer High Sensitivity DNA chip (Agilent, Cat. No. 5067–1504). Samples were pooled and sequenced by McDermott Sequencing Core using Illumina's NextSeq500 to yield paired-end (150 bp long) strand-specific reads.

## RNA-seq data analysis

De-multiplexed raw reads were received from McDermott Sequencing Core. Raw reads were first filtered for phred quality and adapters using FASTQC (*FastQC, Babraham Bioinformatics, URL:* https://www.bioinformatics.babraham.ac.uk/projects/fastqc) and Trimmomatic (*Bolger et al., 2014*). Filtered reads were then aligned to the reference mouse genome mm10 (https://genome.ucsc.edu) using STAR (*Dobin et al., 2013*) aligner. Uniquely mapped reads were used to obtain the gene counts using HTSeq package (*Anders et al., 2015*), and the read counts were normalized using the CPM (counts per million) method implemented in the edgeR package (*McCarthy et al., 2012*; *Robinson et al., 2010*). For further analysis, we performed a sample-specific CPM filtering, considering genes with CPM values of 1.0 or greater in all replicates in one or more of the sleep conditions. DESeq2 (*Anders and Huber, 2010*; *Love et al., 2014*) was used to detect the DEGs across sleep states. We applied a filter of an adjusted *p*-value of $\leq 0.05$ and absolute log fold change of $\geq 0.3$ to identify DEGs. Significant DEGs are visualized using volcano plots. UpSetR (*Conway et al., 2017*) R package was used to identify shared and unique sets of DEGs across sleep states.

## GO analysis

GO analysis of the significant, sleep-states shared and unique DEGs was performed using *ToppGene* (https://toppgene.cchmc.org) (*Chen et al., 2009*) and GO terms were reduced using *REVIGO* (*Supek et al., 2011*). GO categories were considered significant if they contained at least three genes and if they had a Benjamini–Hochberg (B–H)-corrected p-value of $\leq 0.05$.

## DEG enrichment against cell types, MEF2C targets and rPRGs

The mouse visual cortex single cell RNA-seq dataset was downloaded from NCBI Gene Expression Omnibus (accession no. GSE102827) and processed using *Seurat* (*Butler et al., 2018*). Retaining cluster annotations from the original source (*Hrvatin et al., 2018*), genes enriched in each cell-type/cluster were identified using the

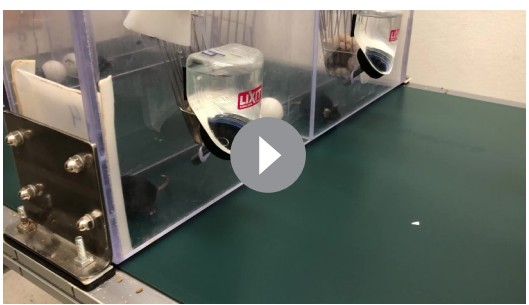

**Video 1.** Mouse on moving treadmill in SD configuration.
https://elifesciences.org/articles/58331#video1

*FindAllMarkers* function of *Seurat* (*Butler et al., 2018*) with default parameters. The list of MEF2C target genes was obtained from *Harrington et al., 2016*. The list of rapid primary response genes was obtained from *Tyssowski et al., 2018*. The list of differentially expressed genes from *Mef2c$^{+/-}$* compared to *Mef2c$^{+/+}$* at CS was obtained from *Harrington et al., 2020*. Enrichment analyses for shared and unique list of DEGs across sleep states against (i) cell types from mouse visual cortex single-cell datasets, (ii) MEF2C targets and (iii) rPRGs were performed using a Fisher's exact test. Also, enrichment analysis for differentially expressed genes across *Mef2c$^{f/f}$* and Mef2c-cKO$^{Camk2a-Cre}$ under CS and differentially expressed genes from *Mef2c$^{+/-}$* compared to *Mef2c$^{+/+}$* at CS was performed using a Fisher's exact test.

## Weighted gene co-expression network analysis

Weighted gene co-expression network analysis (WGCNA) was performed on 19 total RNA-seq samples, including six control sleep (CS) samples, six sleep deprived (SD) and seven recovery sleep (RS) samples. The R package for WGCNA (*Langfelder and Horvath, 2008*) was used to build a gene co-expression network using filtered CPM data (CPM >= 1 across all replicates of one or more of each of the conditions). A signed network was constructed using the *blockwiseModules* function of the WGCNA R package. A value of 26 was chosen as Beta with highest scale-free R square ($R^2$ = 0.796). For other parameters, we used *corType = pearson*, *maxBlockSize = 15000*, *reassignThreshold = 1 × 10$^{-5}$*, *mergeCutHeight = 0.05*, *deepSplit = 4*, *detectCutHeight = 0.999* and *minModuleSize = 50*. Visualizations of network plots were created using Cytoscape v3.4.0 (*Shannon et al., 2003*), representing the top 250 edges based on ranked weights.

## Heat map plot for enrichment of gene lists against WGCNA module genes

List of genes per WGCNA module were enriched against (i) DEGs per sleep state, (ii) cell type genes identified in mouse cortex (*Hrvatin et al., 2018*), (iii) MEF2C targets (*Harrington et al., 2016*), and (iv) module eigengene correlation to each sleep state using *supertest* function of *SuperExactTest* R package. The heat map is generated using *ggplot2* functions *geom_tile*. The modules are ordered based on module eigengene correlation to CS sleep state starting from highest to lowest correlation.

## Down-sampling and permutation DEG analysis

DESeq2 (*Anders and Huber, 2010*; *Love et al., 2014*) differential expression analysis for down-sampled WT CS (n = 4) and SD (n = 3) was performed by first randomly down-sampling the WT CS and SD sample sizes to match the sample sizes of *Mef2c$^{f/f}$*. Down-sampling was performed for 100 permutations and number of DEGs were reported for all 100 permutations.

## Assessment of *Mef2c* exon 11 encoding DNA-binding portion of MEF2C in WT, *Mef2c$^{f/f}$*, Mef2c-cKO$^{Camk2a-Cre}$ mice

Differential exon usage for exons across *Mef2c* was accessed using DEXSeq (*Anders et al., 2012*) indicating only minor variability in exon expression between WT, *Mef2c$^{f/f}$* and Mef2c-cKO$^{Camk2a-Cre}$ samples. Analysis also shows the specific reduction of floxed exon (Exon 11) in Mef2c-cKO$^{Camk2a-Cre}$ mice (*Figure 4*). As expected, this reduction was only partial due to the selective expression of *Camk2a-Cre* that limited CRE-mediated recombination to CAMK-expressing cells.

## Immunoprecipitation of MEF2C protein

Frozen brain tissue from the frontal cortex of mice were immediately homogenized by sonication after addition of 600 mL mRIPA buffer with inhibitors (50 mM Tris-HCl pH 7.5, 150 mM NaCl, 1 mM EDTA, 1% iGEPAL CA-630, 0.5% sodium deoxycholate, 0.1% SDS, 1 mM sodium fluoride, 1 μM cyclosporine A, 1 mM sodium orthovanadate, 50 nM okadaic acid, 1 tablet of cOmplete, EDTA-free Protease Inhibitor Cocktail (Sigma/Roche 11836170001) in deionized water for 10 mL total volume of solution). Samples were then centrifuged at max speed and supernatant was aliquoted. Protein concentration was quantified via Bio-Rad DC Protein Assay (Bio-Rad 5000112). Protein lysates (1.5 mg) were diluted in mRIPA buffer with inhibitors to a total volume of 500 mL. A volume of 2 μL of mouse monoclonal anti-MEF2C antibody (Novus NBP2-00493) was added to each diluted protein

lysate and incubated for 2 hr at 4°C. Samples were then added to 75 µL washed protein G Plus/A agarose bead slurry (Millpore IP05) and incubated for 1 hr at 4°C. The samples were then washed three times with mRIPA buffer with inhibitors and eluted with 2X sample buffer + 10% 2-mercaptoethanol.

## Western blot of MEF2C immunoprecipitation

Immunoprecitation (IP), pre-IP, and post-IP samples were loaded into 7.5% Mini-PROTEAN TGX Stain-Free gels (Bio-Rad 4568026). Gel electrophoresis was run at 250 V for 25 min in 1X Tris-gycine SDS buffer. Proteins were then transferred onto PVDF membranes using the Bio-Rad Trans-Blot Turbo System on the high-MW setting. Membranes were blocked in 1:1 Odyssey blocking buffer/1X PBS for 1 hr. Membranes were incubated overnight with primary antibodies: rabbit anti-phospho-MEF2 (1:200, lab purified) or rabbit anti-MEF2C (1:2500, Abcam ab197070). After washing, Licor secondary antibodies were incubated on the membranes (1:20,000, Licor IRDye 800CW goat anti-rabbit 926–32211). After washing, blots were imaged on the Licor Odyssey CLx system. The pre-IP sample membranes were then incubated with rabbit anti-beta-actin antibody (1:1000, Cell Signaling 4970S) overnight. After washing, the membrane was incubated with the Licor goat anti-rabbit secondary and imaged.

## Analysis of MEF2C immunoprecipitation

Licor image files were imported into Licor ImageStudio software and protein bands were quantified. We calculated a ratio for phospho-MEF2C signal to the total MEF2C signal for each sample. The average ratio for the control samples was used as a normalization factor for each sample. The ratio for each sample were then divided by the normalization factor to generate normalized signal values for each sample. Similar analysis was used to compare total MEF2C signal to beta-actin signal. One-way ANOVA statistical analyses were performed in Graphpad Prism software.

## Brain slice electrophysiological recordings

Brain slices 300 µm thick, cut 15°-angled in rostral direction to keep perpendicular blade penetration in the anterior cingulate area, were prepared from 8- to 12-week-old control (Mef2c fl/fl: Cre-) and Mef2c-cKO$^{Camk2a-Cre}$ mice at the same Circadian time ZT6 immediately after subjecting to one of the three 6 hr sleep/wake protocols (CS, SD, RS). The dissection was done in ice-cold solution containing (mM): NaCl 84, KCl 3, NaH$_2$PO$_4$ 1.25, CaCl$_2$ 0.5, MgSO$_4$ 7, NaHCO$_3$ 26, glucose 20, sucrose 70, ascorbic acid 1.3, kynurenic acid 2 (brought to pH 7.3 by continuous saturation with carbogen gas containing 95% of O$_2$ and 5% of CO$_2$). After dissection, slices were transferred to the carbogen gas-saturated artificial cerebrospinal fluid (ACSF) solution containing (mM): NaCl 125, KCl 5, NaH$_2$PO$_4$ 1.25, CaCl$_2$ 2, MgCl$_2$ 1.3, NaHCO$_3$ 25, glucose 12 (pH 7.3) at 32° C, and were used for experiments after 1 hr equilibration.

Slices were placed in 400 µL recording chamber and continuously perfused at 2 mL/min with heated (33° C) carbogen gas-saturated ACSF containing 1 µm of adenosine receptor 1 blocker 8-cyclopentyl theophylline (CPT) during entire experiment. Solutions containing additionally 100 µm of picrotoxin (PTX) or 1 µl tetrodotoxin (TTX) were washed-in to block GABA receptor synaptic currents during paired-pulse stimulation experiments, and Na$^+$ voltage-dependent channel-generated spontaneous presynaptic spikes during recordings of miniature excitatory post-synaptic currents (mEPSC), respectively.

Pyramidal neurons in cortical layers 2/3 at the anterior cingulate cortex (ACC) area were visually identified with upright microscope equipped with infra-red light sensitive camera, and voltage-clamped at −70 mV holding potential (Cl$^-$ equilibrium potential in our solutions) with patch pipette containing (mM): K-methanesulfonate 130, NaCl 5, MgCl$_2$ 2, Hepes 10, EGTA 1, CaCl$_2$ 0.4 (free Ca$^{2+}$ ~80 nM), MgATP 2, Na$_2$GTP 0.2 (pH ~ 7.3 adjusted with NaOH). Multiclamp 700A amplifier, Digidata 1440A and pClamp 10.2 software (all Molecular Devices, CA) were used for patch-clamp recordings both in voltage-clamp and current clamp modes. The identity of each pyramidal neuron was confirmed by recording a specific pattern of trains of action potentials in current-clamp mode, clearly different from fast-spiking activity of interneurons in response to a series of depolarizing current steps.

CaCl$_2$ was removed, and 0.25 mM of QX 314, a Na$^+$ voltage-dependent channel blocker was added to the patch pipette solution to suppress generation of postsynaptic Na$^+$ currents during recordings of evoked excitatory post-synaptic current (eEPSC) in paired-pulse stimulation experiments. In these experiments, the identity of pyramidal neurons was verified immediately after breaking the membrane patch before suppression of action potentials. Stimulation of axons forming synapses at apical dendrites of L2/3 pyramidal cells during paired pulse experiments was done by placing a bipolar stimulation electrode in the L1 area at a distance ~ 100–200 μm from the recorded neuron axial projection to this area. Brief 0.4 ms biphasic stimulation current pulses of 30 to 60 μA delivered with Biphasic Stimulus Isolator BSI-950 (Dagan, MN) were enough to produce minimal-amplitude reproduceable eEPSCs in 95% of experiments.

## Analysis of *mEPSC*

Functional parameters of miniature excitatory post-synaptic currents (mEPSCs) were calculated from 2 to 4 min continuous recordings following steady-state after application of 1 μl TTX. Recordings were high-pass filtered at 10 Hz and low-pass filtered at 500 Hz. Individual mEPSC were detected by the event template detection algorithm embedded into the pClamp 10 software. A combination of five mEPSCs templates varying from 6 to 25 ms durations was used, each template recognition window being 'trained' from ~ 100 real events. No minimal threshold for mEPSC amplitude was applied for medium and longer events to avoid potential exclusion of low-amplitude attenuated mEPSCs originating from distal dendrites. The same template recognition set of parameters was applied for the analysis of data obtained from all cells included in this study. The microscopic parameter values of individual events (peak amplitude, area, instant frequency) were calculated by the pClamp event analysis algorithm and transferred to the Microsoft Excel for secondary off-line general statistics analysis. Standard box and whiskers graphs for each analyzed mEPSC parameter averaged for all cells within a particular experimental group were plotted using GraphPad Prism software, which represents 25% to 75% interval as vertical boxes, the whole range of data points as whiskers, the line inside boxes as a median, and a '+' sign inside boxes indicates the mean value. One-way ANOVA analysis with Tukey's multiple comparisons test was used to define significant differences within three experimental groups (CS, SD, RS for each genotype) for each reported parameter.

Cumulative histograms for each measured mEPSC parameter of a particular experiment were generated from a continuous 2 to 4 min recording and comprised values falling within a mean ± 3 SD interval (99.7% of events). Within a particular experimental group (CS, SD, RS for each genotype), histograms from individual cells were binned at equal 0.01 steps and then averaged for each binning point over the entire Y-scale range (from 0 to 1). Averaged cumulative histograms were plotted using GraphPad Prism and included mean ± SEM (X-axis error bars) for each binning point. Ordinary one-way ANOVA analysis with Tukey's multiple comparisons test was used to define significant difference between resultant averaged histograms within each genotype group.

Paired-pulse eEPSC analysis. For each paired-pulse experiment, peak amplitudes of coupled evoked EPSCs (eEPSCs) were obtained after averaging traces of 5 to 10 repetitive paired stimulation sequences separated by 20 s relaxation time. P2/P1 amplitude ratio was calculated for three inter-pulse intervals: 20, 50 and 100 ms. Standard box and whiskers graphs for each P2/P1 ratio averaged for all cells within a particular experimental group were plotted for each of three interpulse intervals using GraphPad Prism software. Ordinary one-way ANOVA analysis with Tukey's multiple comparisons test was used to define significant differences within three experimental groups (CS, SD, RS for each genotype) for each measured P2/P1 ratio.

## EEG and EMG acquisition for sleep/waking state analysis

Animals (Mef2c-cKO$^{Camk2a-Cre}$ n = 7, *Mef2c*$^{f/f}$n = 11, C57BL/6 n = 6, *Mef2c*$^{+/-}$ n = 6, *Mef2c*$^{+/+}$ n = 6) were implanted with EEG and EMG electrodes using standard procedures (*Bjorness et al., 2016*). Briefly, mice were anesthetized with isoflurane and placed into a stereotaxic apparatus after which the hair is sheared and the scalp cleaned and incised. Four small holes were drilled for the placement of bilateral skull screw style electrodes over the frontal cortex (AP +1.7, ML +/- 1.77), parietal cortex (AP −1.7, ML +2.0), and occipital cortex (AP −5.5, ML −1.5). Paddle style EMG electrodes (Plastics One) were placed under the dorsal nuchal muscle. Electrode pins were gathered into a six pin pedestal (Plastics One) which was cemented to the skull using dental cement. The wound was closed

using absorbable sutures and coated with triple antibiotic ointment. Mice received buprenorphine for analgesia and were given 14 days to recover from surgical procedures prior to experimental procedures.

Mice were placed into individual cages suspended above the belt of a custom treadmill apparatus with food and water available ad libitum. Mice were given 2 weeks to acclimate to the new environment with daily short duration exposure to the treadmill belt moving after which mice were connected to an amplifier system via a tether attached to the pedestal with an additional week of acclimation to the tether. A balance bar was set between the tether and commutator to allow for unrestricted movement. Following acclimation, EEG and EMG signals were acquired using a 15LT amplifier system (Natus Neurology) for 2 days under baseline (undisturbed) conditions followed by 2 days of chronic, partial sleep deprivation (*Bjorness et al., 2016*). Sleep deprivation consisted of eight cycles of 4 hr TM 'ON' (belt moving) followed by 2 hr TM 'OFF' (belt fixed). State assignment was determined via a custom Matlab (Mathworks) autosorter program using standard criteria (*Bjorness et al., 2016*) for waking, SWS, and REM after which files were manually checked to ensure correct assignment of state and flag epochs featuring artifact. Epochs were scored in 10 s bins with artifact-flagged epochs excluded from power spectral analysis. Episodes began with 30 consecutive seconds of one state and ended with 30 consecutive seconds of a different state. Power spectrum values were calculated in a 2 s window with a 1 s overlap and a Hamming window using the mean squared spectrum function in Matlab. For statistical analyses, Mef2c-cKO$^{Camk2a-Cre}$ was compared to *Mef2c*$^{f/f}$ and C57BL/6, while *Mef2c*$^{+/-}$ was compared to *Mef2c*$^{+/+}$.

### Decay of SWA across the light (inactive) phase

SWA (0.5–4.5 Hz) was normalized to the 24 hr average SWA and the percentage change between the first and last consolidated 1 hr sleep period was calculated as previously described (*Nelson et al., 2013*). The baseline day with the clearest consolidated sleep periods early and late in the light phase were used for statistical comparison via a one-way ANOVA with Sidak's multiple comparison test.

### Decay of SWS-SWA within an averaged SWS episode under baseline conditions

For a full description of the decay analysis see *Bjorness et al., 2016*. Briefly, SWS-SWA power was normalized to W SWA power after which SWS episodes of at least 5 min in duration were collected and gathered into an excel file with one episode per column such that the normalized SWS-SWA power of the first epoch was in the first row of each column, the normalized SWS-SWA power of the second epoch was in the second row of each column, and so forth. Based on the 5 min duration criteria, all columns had at least 30 rows; the longest four episodes were truncated to leave five episodes at the longest time point. One Mef2c-cKO$^{Camk2a-Cre}$, one *Mef2c*$^{f/f}$, and one *Mef2c*$^{+/+}$ mouse per group was excluded on the basis of insufficient episodes of at least 12 min in duration (*Bjorness et al., 2016*). The natural log was taken after which episodes were averaged within animal to create a single averaged SWS episode per mouse followed by averaging across animals within group to result in a single episode per group. Next, the group average episode was transformed by taking the exponential. A fragment of the average was created (excluding the initial ascending phase and periods of high variability toward the end) and fit using a single phase decay (GraphPad Prism) in order to determine the time constant of decay ($\tau$). Based on previously described exclusion criteria (absolute sum of squares, *Bjorness et al., 2016*), the averaged group episode for Mef2c-cKO$^{Camk2a-Cre}$ mice could not be fit. Thus, the time constant of decay determined for the *Mef2c*$^{f/f}$ group was used to determine the plateau and R$^2$ value for the Mef2c-cKO$^{Camk2a-Cre}$ group.

### Rebound SWS-SWA following chronic, partial SD

Mice underwent 8 cycles of 4 hr SD (TM 'ON') followed by 2 hr RS (TM 'OFF') starting at ZT0. SWS-SWA power was averaged in 2 hr bins. Rebound SWS-SWA was determined by the percent change in SWS-SWA from baseline conditions to TM 'OFF' conditions using time-matched circadian bins (i.e. ZT 4–6, ZT 10–12, ZT 16–18, ZT 22–24). A one-way ANOVA was used to compare average rebound across the 8 RS periods and a One sample T test was used to compare average values to a theoretical mean of 0 (GraphPad Prism).

## Time in sleep/waking state

Time in each sleep/waking state (waking, SWS, REM) under baseline conditions was averaged in 2 hr bins and calculated as the percent time of the total period. Averaged percent time in state for each state was compared between groups using a one-way ANOVA (GraphPad, Prism).

## Cumulative distribution of SWS episodes

SWS episodes under baseline and SD (recovery periods) conditions were binned by duration using the histogram function in excel (Microsoft); the first bin was 90 s with successive bins increasing by 40 s. The number of episodes in each bin was divided by the total number of episodes for the condition and multiplied by 100 to get a percentage of the total episodes represented by each bin. Next, a cumulative percentage was calculated by adding the percentage of successive bins. A two-way ANOVA (condition, bin) was used to compare cumulative histogram episode durations across groups and conditions (baseline and SD).

## Spectral power distribution across state

Under baseline conditions, individual 10 s epochs were separated by state (waking, SWS, REM) after which spectral power in each 1 Hz bin up to 50 Hz was normalized by the total power for that state (waking epoch normalized to waking total power, etc). Fraction power was then averaged across epochs within each 1 Hz bin to get an average spectral distribution from 1 to 50 Hz by state. Two-way ANOVA (group, frequency) was used to compare spectral distributions by state across groups.

## Acknowledgements

We thank Mr. To Thai, Ms. Lilian Zhan and Dr. Busra Goksu for technical assistance and Dr. Stefano Berto for discussions about genomic analyses. Funding: This study was funded by NIH grant R01 MH080297 and WPI program for IIIS to RWG; NIH grants RO1 MH102603, DC014702 and the James S McDonnell Foundation 21$^{st}$ Century Science Initiative in Understanding Human Cognition – Scholar Award (220020467) to GK; NIH grants R01 AG045795, R01 NS 106657 to JST; JST is an Investigator in the Howard Hughes Medical Institute; NIH grant RO1 MH111464 to CWC and F30 HD098893 to CMB. The contents do not represent the views of the U.S. Department of Veterans Affairs or the United States Government.

## Additional information

### Competing interests

Genevieve Konopka: Reviewing editor, *eLife*. The other authors declare that no competing interests exist.

### Funding

| Funder | Grant reference number | Author |
| --- | --- | --- |
| National Institute of Neurological Disorders and Stroke | NS103422 | Robert W Greene |
| National Institute of Mental Health | MH102603 | Robert W Greene |
| National Institute on Deafness and Other Communication Disorders | DC014702 | Genevieve Konopka |
| James S. McDonnell Foundation | 220020467 | Genevieve Konopka |
| National Institute on Aging | AG045795 | Joseph S Takahashi |
| National Institute of Neurological Disorders and Stroke | NS106657 | Joseph S Takahashi |

| Howard Hughes Medical Institute | | Joseph S Takahashi |
| National Institute of Mental Health | MH111464 | Christopher W Cowan |
| Eunice Kennedy Shriver National Institute of Child Health and Human Development | HD098893 | Catherine Bridges |
| International Institute for Integrative Sleep | | Robert W Greene |
| North Texas Veterans Affairs Health Care System for Research | | Theresa E Bjorness |

The funders had no role in study design, data collection and interpretation, or the decision to submit the work for publication.

### Author contributions

Theresa E Bjorness, Ashwinikumar Kulkarni, Volodymyr Rybalchenko, Formal analysis, Investigation, Writing - original draft, Writing - review and editing; Ayako Suzuki, Investigation, Writing - review and editing; Catherine Bridges, Adam J Harrington, Formal analysis, Investigation, Writing - review and editing; Christopher W Cowan, Genevieve Konopka, Supervision, Writing - original draft, Writing - review and editing; Joseph S Takahashi, Supervision, Writing - review and editing; Robert W Greene, Conceptualization, Supervision, Funding acquisition, Writing - original draft, Writing - review and editing

### Author ORCIDs

Ashwinikumar Kulkarni (iD) http://orcid.org/0000-0003-0951-2427
Christopher W Cowan (iD) http://orcid.org/0000-0001-5472-3296
Joseph S Takahashi (iD) http://orcid.org/0000-0003-0384-8878
Genevieve Konopka (iD) https://orcid.org/0000-0002-3363-7302
Robert W Greene (iD) https://orcid.org/0000-0003-1355-9797

### Ethics

Animal experimentation: This study was performed in strict accordance with the recommendations in the Guide for the Care and Use of Laboratory Animals of the National Institutes of Health. All of the animals were handled according to approved institutional animal care and use committee (IACUC) protocols of the UT Southwestern Medical Center and the North Texas VA Health Care System. All surgery was performed under isoflurane anesthesia, and every effort was made to minimize suffering.

### Decision letter and Author response

Decision letter https://doi.org/10.7554/eLife.58331.sa1
Author response https://doi.org/10.7554/eLife.58331.sa2

## Additional files

### Supplementary files

• Supplementary file 1. Differentially expressed genes (DEG) across sleep states. (1) DEG for CS to SD, (2) DEG for CS to RS, (3) DEG for SD to RS, (4) list of overlapping and unique genes across sleep states (*Figure 1D*) and (5) list of overlapping genes between genes shared by SD and RS against rPRGs (Tyssowski et al.) and ERTFs (Hrvatin et al.)

• Supplementary file 2. Overlapping SD genes with glucocorticoid cellular response genes. The GC response genes are a subset (27/44) from MGI GO:0071385 that we identified as being expressed in the frontal cortex.

• Supplementary file 3. GO for shared and unique differentially expressed genes (DEG). (1) GO for genes shared by SD and RS with increased expression against CS, (2) GO for genes shared by SD and RS with decreased expression against CS, (3) GO for genes unique to SD with increased expression against CS, (4) GO for genes unique to SD with decreased expression against CS, (5) GO for genes unique to RS with increased expression against CS and (6) GO for genes unique to RS with decreased expression against CS.

• Supplementary file 4. GO for gene modules identified by weighted correlation network analysis (WGCNA). (1) Intermodular connectivity across modules, (2) genes associated with each module (3-61) GO for genes within each module.

• Supplementary file 5. Differentially expressed genes (DEG) across sleep states for Mef2c-cKO-$^{Camk2a-Cre}$. (1) DEG for CS to SD for $Mef2c^{f/f}$ and (2) DEG for CS to SD for Mef2c-cKO$^{Camk2a-Cre}$.

• Supplementary file 6. Functional parameters of mEPSCs. Recordings were obtained ex Vivo from anterior cingulate cortex excitatory neurons of $Mef2c^{f/f}$ and Mef2c-cKO$^{Camk2a-Cre}$ mice exposed to three different sleep/wake experimental conditions: control sleep 6 hr (CS), sleep deprivation 6 hr (SD) and sleep deprivation 4 hr followed by recovery sleep 2 hr (RS).

• Supplementary file 7. Paired pulse ratio of evoked EPSCs. $P_2/P_1$ was obtained at three different interpulse intervals (20, 50 and 100 ms), obtained in anterior cingulate cortex excitatory neurons of $Mef2c^{f/f}$ and Mef2c-cKO$^{Camk2a-Cre}$ mice exposed to three different sleep/wake experimental conditions: control sleep 6 hr (CS), sleep deprivation 6 hr (SD) and sleep deprivation 4 hr followed by recovery sleep 2 hr (RS).

• Supplementary file 8. Overlapping genes shared by (a) differentially expressed genes across $Mef2c^{f/f}$ and Mef2c-cKO$^{Camk2a-Cre}$ at CS and (b) differentially expressed genes across $Mef2c$-Hets and CTL at CS.

• Supplementary file 9. Number of detected DEGs for CS (n = 4) and SD (n = 3) in random downsampling and permutation experiment over 100 iterations in comparison to WT DEGs for CS (n = 6) and SD (n = 7).

• Supplementary file 10. Tab 'Overlap': Overlap between DEGs for WT_CS|SD (6248 genes) and Mef2cff_CS|SD (767 genes). Tab 'Gene Ontology: GO of overlapped genes, *ToppGene* (https://top-pgene.cchmc.org).

• Transparent reporting form

## Data availability

The NCBI Gene Expression Omnibus (GEO) accession number for the RNA-seq data reported in this paper is GSE 144957.

The following dataset was generated:

| Author(s) | Year | Dataset title | Dataset URL | Database and Identifier |
|---|---|---|---|---|
| Bjorness TE, Kulkarni A, Rybalchenko V, Suzuki A, Bridges C, Cowan CW, Takahashi JS, Konopka G, Greene RW | 2020 | An essential role for MEF2C in the cortical response to loss of sleep | http://www.ncbi.nlm.nih.gov/geo/query/acc.cgi?acc=GSE144957 | NCBI Gene Expression Omnibus, GSE144957 |

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
