## [Decision Letter]

**Acceptance summary:**

Sleep abnormalities are associated with long-term changes in brain function. The authors of this study investigated the transcriptional changes in frontal cortex that accompany sleep loss in mice in order to gain insight into the molecular basis of these plasticity changes. The key results reveal transcriptional mechanisms that underlie changes in brain function with sleep deprivation and strong functional evidence of a role for MEF2C in the regulation of synaptic plasticity and sleep homeostasis.

**Decision letter after peer review:**

Thank you for submitting your article "An essential role for MEF2C in the cortical response to loss of sleep" for consideration by *eLife*. Your article has been reviewed by three peer reviewers, one of whom is a member of our Board of Reviewing Editors, and the evaluation has been overseen by Catherine Dulac as the Senior Editor. The following individual involved in review of your submission has agreed to reveal their identity: Graham Diering (Reviewer #2).

The reviewers have discussed the reviews with one another and the Reviewing Editor has drafted this decision to help you prepare a revised submission.

Summary:

Sleep abnormalities are associated with long-term changes in brain function. The authors of this interesting and well-written study investigated the transcriptional changes in frontal cortex that accompany sleep loss in mice in order to gain insight into the molecular basis of these plasticity changes. The study is very well done including controls for circadian effects, and the key results are striking, with an interesting investigation of the transcriptional mechanisms that underlie changes in brain function with sleep deprivation and strong functional evidence of a role for MEF2C in the regulation of synaptic plasticity and sleep homeostasis.

There are two issues that need to be addressed by the authors with text revisions.

Essential revisions:

1) The reviewers were generally impressed by the transcriptomic data, however they raised two specific concerns that need to be addressed.

– Why do we see 767 DEGs in the *Mef2C*^f/f^ control mice in Figure 3, compared with 6248 DEGs in the WT mice of Figure 1, with roughly the same number of genes (11-13k) detected? This was confounding to all three of the reviewers and requires a clear, direct explanation.

– While the authors would like to make the point that many transcripts continue to be mis-expressed even after RS, their study wasn't really designed to address this. Comparing DEGs across the different conditions examined by the authors, certainly, the difference between 6248 (or 3827) and 81 is massive. However, it is unclear how much of this is due to insufficient recovery sleep under their conditions, especially considering that many transcripts may have longer half-lives that would make effects of a 2h "recovery" difficult to quantify accurately. This conclusion might better be made with methods to look at transcription initiation per se, and a fairer way of examining these data would be to ask how many of the 6248 genes changed expression levels toward baseline, significantly or not. This is not the main point of the manuscript and the discussion of this section could certainly be softened, however if retained, stronger and considerable qualification of the arguments presented would be needed.

2) The functional studies on MEF2C are deep and compelling, however MEF2C does not leap out of the transcriptomic analyses the way the authors seem to suggest in the text. The reviews proposed several text revisions to the MEF2C section to make it stronger.

– Mef2C rather "drops out of the blue", first at the end of the Introduction after a single sentence, and then appears in the Results without much justification (the authors suggest the gene expression data "pointed them" to Mef2C in unbiased fashion, but it doesn't really). Discussing Harrington et al. more fully in the Introduction would greatly aid the reader in following and appreciating this story.

– The authors need to back off or better support the text suggesting the transcriptomic data point to MEF2C.

One reviewer noted, "It starts with Figure 1E: downregulated DEGs shared by SD and RS were enriched for neuronal genes downregulated by *Mef2c* loss of function. The statistics to me are unclear here. What is the actual overlap? The general nature of Figure 1E does not allow the reader to evaluate this conclusion very well. Then it looks like Figure 2 is showing exactly the opposite of what its title suggests. Very few WGCNA modules overlap between the SD analysis and the set of genes known to be regulated by Mef2C. Much of the data in this paper is striking, but this figure is unconvincing."

A second version of this concern raised by another reviewer suggested removing the Emx1-Cre MEF2C cKO data comparisons from the manuscript to solve this problem. The reviewer said, "Considering only the data presented from the CamKII-Cre cKO mice, the data are highly consistent showing a blunting of the transcriptional responses, the synaptic responses and the EEG responses to SD in the cKO compared with f/f mice. (The text indicates that both gene expression and synaptic function differ in the control condition between f/f and cKO mice and the authors suggest that there may be homeostatic compensations in the cKO mice prior to SD that contribute to the blunted response, which is an interesting model). However, the paper also tries to tell their MEF2C story using a previous set of RNAseq data from Emx-Cre MEF2C cKO mice. The pattern of knockout in the Emx1-Cre and CamKII-Cre lines is not the same and functional data are available here only for CamKII-Cre. The Emx1-Cre RNAseq data are counterproductive and their use to motivate the further study of MEF2C feels forced. I would pull these data."

– All three reviewers agreed that the MEF2C phosphorylation data are not compelling and should be removed. The relationship between the phosphorylation and the activation of MEF2C is complex, with both concerted dephosphorylation (as seen by the downshift on western) and specific increases in phosphorylation (Thr 293/300/Ser 387, which would require phospho-antibodies to see) associated with activation. The reviewers as a group were confused by the text about whether the authors were proposing that MEF2C activity was increased or decreased during SD (which depends in part on whether the authors think the relevant functions of MEF2C in this context are as an activator or a repressor, which also was differentially interpreted by the reviewers). Anything the authors can do to clarify their model of MEF2C action in SD would benefit the text, but removing the phosphorylation from Figure 3 including the cartoon will especially benefit this part of the story.

– Finally, one reviewer recommended the analysis of the MEF2C germline HET mice should be brought up to the main figures rather than relegated to the supplementary and the comparison between the HET and the CamKII-Cre mice should be more completely discussed. One particularly interesting issue that is briefly mentioned but never addressed in the context of the HET mice is the possibility that MEF2C may have both neuronal and non-neuronal functions.

---

## [Author Response]

Essential revisions:1) The reviewers were generally impressed by the transcriptomic data, however they raised two specific concerns that need to be addressed.– Why do we see 767 DEGs in the Mef2C^f/f^ control mice in Figure 3, compared with 6248 DEGs in the WT mice of Figure 1, with roughly the same number of genes (11-13k) detected? This was confounding to all three of the reviewers and requires a clear, direct explanation.

We agree with the reviewers that at first glance, this is an unexpected result. It is perhaps even more surprising given the fact that mEPSCs and EEG indicators of sleep need are not different between WT and floxed mice (see Figure 6 and Figure 4—figure supplement 1).

We have added the following into the text of the manuscript in hopes of better clarification of this issue:

*“*Although the observed attenuation of *Mef2c*^f/f^ mouse DEGs from CS to SD induced by the conditional loss of *Mef2c* is striking, it is of note that the number of CS to SD DEGs comparing WT to *Mef2c*^f/f^ also shows some attenuation. […] Furthermore, the synaptic and EEG phenotypes of these genotypes across sleep conditions are indistinguishable as shown below (Figure 4—figure supplement 1 and Figure 6).”

Importantly, such differences ultimately do not affect physiological profiles (synaptic strength or EEG measurements of sleep need) inferring that any differences between WT and floxed mice at the transcriptomic level are just “noise” and do not have a functional read out affecting sleep condition-dependent cellular state. Since these responses are absent in the cKO animals, the CS to SD differential expression that is essential for both the similar WT or the *Mef2c^f/f^* sleep SD phenotype requires intact *Mef2c* genes.

– While the authors would like to make the point that many transcripts continue to be mis-expressed even after RS, their study wasn't really designed to address this. Comparing DEGs across the different conditions examined by the authors, certainly, the difference between 6248 (or 3827) and 81 is massive. However, it is unclear how much of this is due to insufficient recovery sleep under their conditions, especially considering that many transcripts may have longer half-lives that would make effects of a 2h "recovery" difficult to quantify accurately. This conclusion might better be made with methods to look at transcription initiation per se, and a fairer way of examining these data would be to ask how many of the 6248 genes changed expression levels toward baseline, significantly or not. This is not the main point of the manuscript and the discussion of this section could certainly be softened, however if retained, stronger and considerable qualification of the arguments presented would be needed.

The point we wanted to make with this result is that the response to SD only slowly recovers during RS (i.e. >2hrs) for a large number of DEGs. The reviewers are correct that we do not examine the recovery period longer than 2hrs. While that will make for a very interesting future study, we hope the reviewers appreciate that is it outside of the scope of this current study. We have now included language that a 2hr time point might be insufficient to see transcripts with longer half-lives potentially return to baseline using the current design:

“We would also like to note that the design of our current study only examined gene expression changes after two hours of recovery sleep. […] Future studies that use additional longer time periods of recovery sleep can examine this possibility.”

Most genes that are significantly changed for CS|SD show non-significant change towards baseline by 2h of RS. Moreover, we plotted a histogram of the number of genes changing based on FDR cut-off (Author response image 1) and we observe few changes in the number of genes changing until ~FDR>=0.75. This suggests that we cannot predict the additional transcriptomic changes that would occur with longer SD using the current dataset.

**Author response image 1. sa2fig1:** 

2) The functional studies on MEF2C are deep and compelling, however MEF2C does not leap out of the transcriptomic analyses the way the authors seem to suggest in the text. The reviews proposed several text revisions to the MEF2C section to make it stronger.– Mef2C rather "drops out of the blue", first at the end of the Introduction after a single sentence, and then appears in the Results without much justification (the authors suggest the gene expression data "pointed them" to Mef2C in unbiased fashion, but it doesn't really). Discussing Harrington et al. more fully in the Introduction would greatly aid the reader in following and appreciating this story.– The authors need to back off or better support the text suggesting the transcriptomic data point to MEF2C.

We thank the reviewers for this suggestion and now more fully discuss the rationale for focusing on MEF2C in the Introduction. We also now make it clear in the Results that we examined overlap of our data with previously published MEF2C-transcriptomic datasets because of the known role for MEF2C in regulating excitatory neuron synaptic strength.

One reviewer noted, "It starts with Figure 1E: downregulated DEGs shared by SD and RS were enriched for neuronal genes downregulated by Mef2c loss of function. The statistics to me are unclear here. What is the actual overlap?

We apologize for omitting the statistical test in the main text and legend and only provided this information in the Materials and methods. A Fisher’s exact test was used to calculate the significance of the overlap of the sleep DEGs and the *Mef2c* DEGs. The size of the bubble indicates the odds ratio for enrichment and the color indicates the p-value. In this case the p-values are 3.060^e-05^ for upregulated genes (102 genes common between shared up-regulated sleep genes and Mef2c-cKO*^Emx1^* up-regulated genes) and 4.116^e-28^ for downregulated genes (181 genes common between shared down-regulated sleep genes and Mef2c-cKO*^Emx1^* down-regulated genes).

The general nature of Figure 1E does not allow the reader to evaluate this conclusion very well.

We hope this description makes the significance of these overlaps clearer.

Then it looks like Figure 2 is showing exactly the opposite of what its title suggests. Very few WGCNA modules overlap between the SD analysis and the set of genes known to be regulated by Mef2C. Much of the data in this paper is striking, but this figure is unconvincing."

We again apologize if this was not clear. The WGCNA represents the distribution of *all* expressed genes in the mouse frontal cortex in our dataset and breaks down these genes into modules of co-expression. One would never expect to find all modules in a WGCNA analysis represented by a particular trait, especially when considering this is a dataset of over 13,000 expressed genes. The fact that we observe significant enrichment for *Mef2c* target genes in 15 out of the 59 modules is remarkable (Monte Carlo P-value: 0.001; now added to text). When all the expressed genes are randomly grouped into modules to simulate the 59 detected modules (controlling for module sizes) over and over again (1000 iterations), the chance of 15 or more modules to show significant enrichment for *Mef2c* target genes is extremely rare (Author response image 2). The list of *Mef2c* target genes is comprised of 1,076 genes. Thus, out of 13,000 expressed genes, 1,005 (441 up-regulated and 564 down-regulated) *Mef2c* target genes are coordinately co-expressed in our dataset of wildtype mice undergoing sleep deprivation into modules. Moreover, these modules are primarily modules represented by sleep-related eigengenes. In other words, the first principal component driving the coordination of expression of the genes in these modules is sleep state.

We now state in the manuscript:

“Notably, putative target genes of MEF2C, were enriched in a surprising, 15 of 59 modules generated from the three sleep conditions employed in this study. […] Furthermore, most of the enriched modules had sleep condition related eigengenes indicating that sleep condition was the principal component factor driving their coordinated expression.”

A second version of this concern raised by another reviewer suggested removing the Emx1-Cre MEF2C cKO data comparisons from the manuscript to solve this problem. The reviewer said, "Considering only the data presented from the CamKII-Cre cKO mice, the data are highly consistent showing a blunting of the transcriptional responses, the synaptic responses and the EEG responses to SD in the cKO compared with f/f mice. (The text indicates that both gene expression and synaptic function differ in the control condition between f/f and cKO mice and the authors suggest that there may be homeostatic compensations in the cKO mice prior to SD that contribute to the blunted response, which is an interesting model). However, the paper also tries to tell their MEF2C story using a previous set of RNAseq data from Emx-Cre MEF2C cKO mice. The pattern of knockout in the Emx1-Cre and CamKII-Cre lines is not the same and functional data are available here only for CamKII-Cre. The Emx1-Cre RNAseq data are counterproductive and their use to motivate the further study of MEF2C feels forced. I would pull these data."

We agree with the reviewers that these models are not exactly the same. However, we would prefer to use the results in Figure 1E and the WGCNA as justification for our examination of the CamKII cKO mice. A postnatal deletion of *Mef2c* seems relatable to sleep studies but we didn’t want the transcriptomics to come out of nowhere. However, if all reviewers and the editors still feel that these data need to be removed, we will do so.

– All three reviewers agreed that the MEF2C phosphorylation data are not compelling and should be removed. The relationship between the phosphorylation and the activation of MEF2C is complex, with both concerted dephosphorylation (as seen by the downshift on western) and specific increases in phosphorylation (Thr 293/300/Ser 387, which would require phospho-antibodies to see) associated with activation.

We have now completely rewritten these results to better clarify them and underscore their significance:

“Because MEF2 transcriptional activity is increased upon activity-dependent dephosphorylation of S396 (Flavell et al., 2006), and expression of a constitutively-active form of MEFC (MEF2C-VP16) is sufficient to reduce cortical excitatory synapses (Harrington, 2016), we speculated that sleep loss might decrease MEF2C phosphorylation at S396. […] In contrast, RS did not significantly alter MEF2C P-S396 levels (Figure 3D), suggesting that SD triggers signaling events to transiently activate MEF2C-dependent transcription and promote the weakening and/or elimination of excitatory synapses.”

We have also rewritten the Introduction section relevant to the phosphorylation of MEF2C as follows:

“Neuronal activity associated with waking, that may include neuronal depolarization, firing and arousal associated neuromodulator activity, can switch MEF2C from a transcriptional repressor to an activator. […] However, the relationship between MEF2C and sleep-regulated transcripts, sleep-regulated excitatory synaptic strength and sleep-homeostatic EEG activity remains unexplored.”

The reviewers as a group were confused by the text about whether the authors were proposing that MEF2C activity was increased or decreased during SD (which depends in part on whether the authors think the relevant functions of MEF2C in this context are as an activator or a repressor, which also was differentially interpreted by the reviewers). Anything the authors can do to clarify their model of MEF2C action in SD would benefit the text, but removing the phosphorylation from Figure 3 including the cartoon will especially benefit this part of the story.

We apologize if this was not clear. MEF2C is activated by SD and this results in both up and down-regulation of direct and indirect target genes that facilitate both excitatory synapse-weakening and strengthening genes, respectively. We have improved Figure 3E to make this point.

– Finally, one reviewer recommended the analysis of the MEF2C germline HET mice should be brought up to the main figures rather than relegated to the supplementary and the comparison between the HET and the CamKII-Cre mice should be more completely discussed. One particularly interesting issue that is briefly mentioned but never addressed in the context of the HET mice is the possibility that MEF2C may have both neuronal and non-neuronal functions.

We have now brought the figure illustrating the analysis of *Mef2c^+/-^* mice into the main figures as Figure 7, as kindly suggested by the reviewers. We have also added the following sentence:

“Moreover, the non-overlapping genes can reflect either developmental effects or non-neuronal effects present in the haploinsufficiency model, but absent in the Mef2c-cKO*^Camk2a-Cre^* mice.”, to briefly note the possibility of both non-neuronal and developmental differences in *Mef2c^+/-^* mice.